# Activation Functions and Normalization in Deep Continual Learning

## Abstract

Deep learning models often struggle to remain adaptable in continual learning scenarios, where the data distribution changes over time. Beyond the well-known challenge of catastrophic forgetting, these models also face plasticity loss that is characterized as the gradual decline in their ability to learn from future data. We study plasticity loss through the lens of activation and normalization interactions. Through a large-scale empirical study, we evaluate 26 activation functions across three normalization strategies using ResNet-18 on the class-incremental CIFAR-100 benchmark. Our findings reveal that plasticity is not determined by any single design choice, but rather is influenced by the complex interaction between activation functions and normalization layers. We uncover a link between overfitting and plasticity loss, and show that simple yet effective training strategies, such as applying soft labels, learning rate warm-up and excluding affine normalization parameters from L2 regularization can significantly slow down the emergence of plasticity loss. Based on these findings, we offer additional recommendations for model design and training, we keep the networks inherently more performant and adaptable over a long time without any active component.

**Keywords:** Activation Functions, Normalization, Continual Learning, Plasticity

## 1 Introduction

Continual learning is a significant challenge for deep neural networks. This boils down to the following key question:

*How is it possible to learn from a stream of data without forgetting past knowledge* or *losing the ability to learn from future data?*

While much of the literature already focuses on *catastrophic forgetting* (McCloskey & Cohen, 1989; French, 1999; Mnih et al., 2013; Goodfellow et al., 2013), the tendency of models to overwrite earlier knowledge, recent publications have brought attention to a more subtle problem: *plasticity loss* (Abbas et al., 2023; Lyle et al., 2024b; Dohare et al., 2024). Unlike forgetting, which emerges as a loss in accuracy on past data, plasticity loss refers to the network's declining ability to learn effectively from future data even when past knowledge is not negatively affected. In this work, we focus on this often overlooked problem. Existing approaches (Kirkpatrick et al., 2017; Dohare et al., 2024; Sokar et al., 2023; Nikishin et al., 2024; Ash & Adams, 2020; Elsayed & Mahmood, 2024) for plasticity loss largely monitor the emergence of decline, and then attempt to fix it after it slowly appears. While these methods have great value, we argue for a *complementary* and *proactive* approach. We think it is necessary to build robust model architectures and use suitable optimization techniques to delay the emergence of plasticity loss for as long as possible. Specifically, we explore how several training routines and core architectural choices, such as activation functions and normalization layers, interact and eventually affect a model's long-term adaptability.

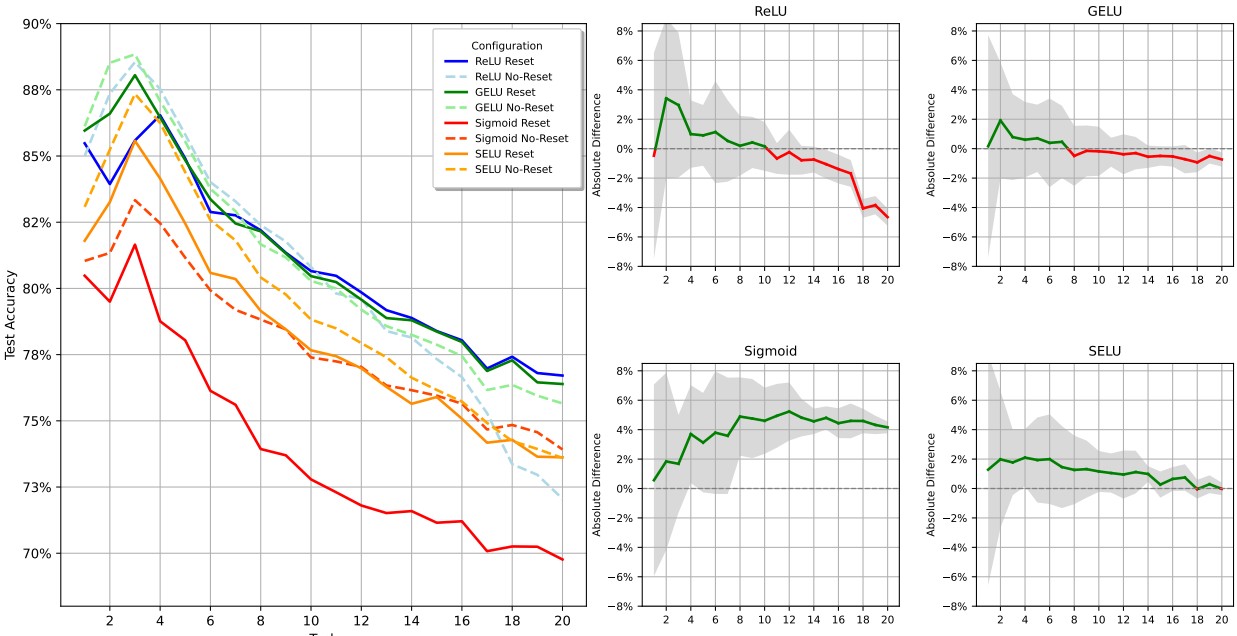

Figure 1: **Patterns of Plasticity Behavior.** This figure visualizes the training behavior of four activation functions when paired with Batch Normalization using class-incremental CIFAR-100. The left panel shows the mean (std. is on the right) test accuracy for *Reset* and *No-Reset* runs, where *Reset* serves as a baseline in which model weights are trained only on one task. The right panels presents the absolute difference in accuracy between the two methodologies, anchored at 0% by using the *Reset* performance. From these comparisons, we identify four characteristic patterns of plasticity. (1) **Fast Decline (RELU)**, where adaptability quickly degrades. (2) **Slow Decline (GELU)**, where performance decreases more gradually. (3) **Improvement (Sigmoid)**, where learning benefits from prior tasks. And (4) **On-Par (SELU)**, where both training protocols perform similarly, indicating minimal to no loss of plasticity.

## 1.1 Performance or Plasticity?

Randomly initialized networks are highly plastic, meaning that they can potentially adapt quickly to unseen data. But obviously with random weights, they perform very poorly on any given task. In contrast, heavily trained networks perform well on their learned objective but often suffer from a decline in plasticity (Goodfellow et al., 2013; Lee et al., 2024a; Khetarpal et al., 2022; Abbas et al., 2023). Again, this raises a central question:

*Is plasticity inevitably sacrificed for performance, or can both be achieved together?*

To answer this, we must revisit assumptions built into model design. While operations like matrix multiplication are universal and shared across architectures (Goodfellow et al., 2016a; Vaswani et al., 2017), components such as activation functions and normalization layers are often treated as interchangeable (He et al., 2016; Liu et al., 2022; Ramachandran et al., 2017; Ioffe, 2015; Ba et al., 2016). Therefore, these components may have an important role in how plasticity emerges and degrades over time. This is particularly relevant for continual learning, where a model's quality depends not just on how well it learns now, but how well it can continue to learn in the future.

In this work, we conduct an empirical study across 26 activation functions, including common choices like ReLU and GELU as well as novel and custom variants, paired with three normalization strategies: BatchNorm, LayerNorm, and no normalization. Using ResNet-18 on the class-incremental CIFAR-100 benchmark, we systematically evaluate how these choices affect plasticity loss over time.

Our findings reveal that the interaction between activation and normalization is far from trivial:

- Some combinations significantly delay plasticity loss and improve robustness,

- Others lead to overfitting and decline in adaptability,

- And no single activation function consistently outperforms across all settings.

## 1.2 Contributions

This work makes the following key contributions:

1. We present a large-scale empirical analysis of the relationship between activation functions, normalization strategies, and plasticity loss in deep continual learning.

2. We create novel activation functions and identify activation-normalization pairs that result in good plasticity and performance.

3. We show that simple training modifications, such as soft labels, learning rate warm-up and selective L2 regularization can significantly improve plasticity.

4. We believe that existing active methods together with proactive approaches like ours offer a more integrated framework for building robust continual learning models.

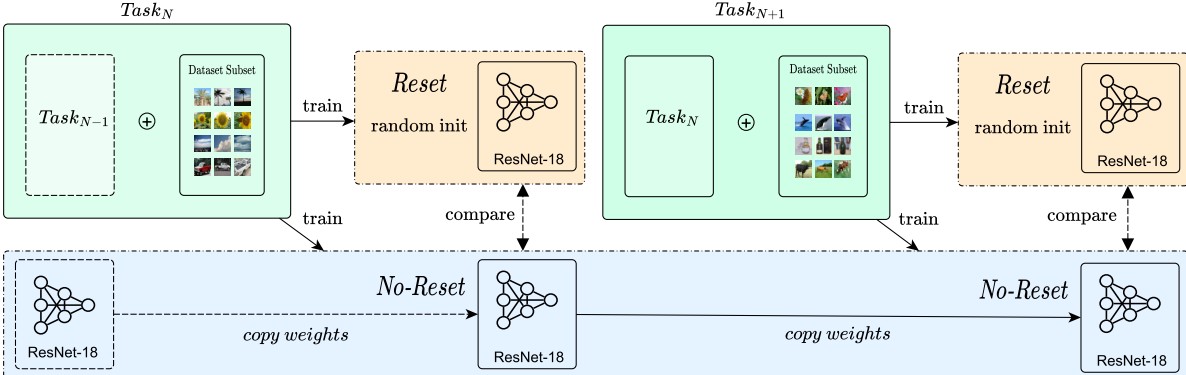

Figure 2: **Schematic of Continual Learning with Classification Datasets.** This diagram explains the class-incremental training setup used in continual learning benchmarks such as CIFAR-100. Training starts with a small subset of the dataset and progressively adds additional classes over time. Each extension of the subset is defined as a "Task", where the model is trained while re-using weights from the previous task (*No-Reset*). Evaluation metrics for quantifying plasticity loss, such as test accuracy, are measured after each task and compared with baseline models (*Reset*) which are only trained on the current task.

**Architecture.** This work specifically focuses on studying residual networks (ResNets) (He et al., 2016) as they are generally more capable than classic convolutional neural networks (CNNs) and demonstrate superior performance in supervised learning (He et al., 2016; Liu et al., 2022; Szegedy et al., 2017; Sandler et al., 2018; Zhao et al., 2024) and continual learning tasks such as reinforcement learning (RL) (Espeholt et al., 2018; Schwarzer et al., 2023; Obando-Ceron et al., 2024). We use ResNet-18 in our experiments (see Figure 2) and build them with Batch Normalization (BatchNorm) (Ioffe, 2015), Layer Normalization (LayerNorm) (Ba et al., 2016) or no normalization layer (No-Norm). Those layers and other alternatives (Bjorck et al., 2021) offer good solutions, but their interaction with different activation functions and the resulting impact on plasticity loss will be studied in this paper.

## 2 Background

The ability of neural networks to learn continually without suffering from plasticity loss or catastrophic forgetting remains a fundamental challenge in deep learning. While catastrophic forgetting has been extensively studied (McCloskey & Cohen, 1989; French, 1999; Mnih et al., 2013; Goodfellow et al., 2013), where previously acquired knowledge is forgotten, a more subtle but important issue is plasticity loss (Lyle et al., 2024b; Dohare et al., 2024; Sokar et al., 2023). This phenomenon describes a network's diminishing ability to adapt to new information over time, even if catastrophic forgetting is addressed explicitly (Mnih et al., 2013). Unlike catastrophic forgetting, plasticity loss does not necessarily degrade past knowledge but rather makes learning increasingly difficult, eventually freezing or collapsing the network's trainability (Lyle et al., 2024a;b).

**Existing solutions** that address plasticity loss have been largely *reactive*, focusing to restore the trainability after signs of degradation appear. For example, Plasticity Injection (Nikishin et al., 2024), Continual Backpropagation (Dohare et al., 2024), and ReDo (Sokar et al., 2023) attempt to revive dormant neurons or pathways. UPGD (Elsayed & Mahmood, 2024) perturbs gradients based on utility, while periodic layer resets (D'Oro et al., 2022; Schwarzer et al., 2023) periodically refresh parts of the model to counter cumulative inflexibility. Proactive strategies remain rarer but promising. Methods like Shrink and Perturb (Ash & Adams, 2020) and the Hare & Tortoise framework (Lee et al., 2024b) use architectural safeguards such as soft resets or slow and fast adapting modules to remain adaptable. However, many of these approaches operate independently of the model's representational components or are even constrained to certain activations. In contrast, our work investigates whether *activation functions* and *normalization techniques*, core elements of a model's architecture, can be cleverly chosen to resist plasticity issues for as long as possible.

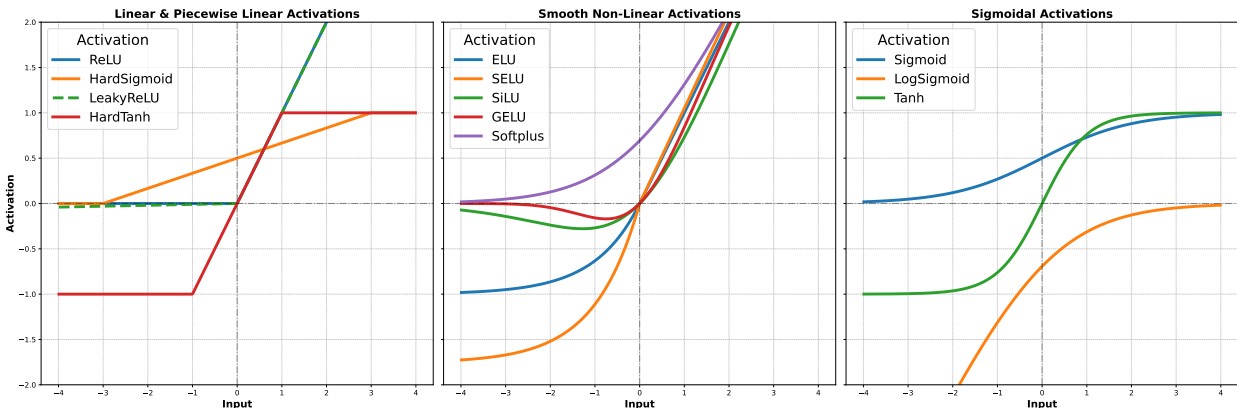

Figure 3: **Overview of Selected Activation Functions.** We present a unified visualization for the first twelve activation functions, grouped into three broad categories. From left to right: **(1)** Piecewise Linear Activations showing sharp transitions, **(2)** Smooth Non-Linear Activations characterized by continuous, differentiable curves, and **(3)** Sigmoidal Activations which show the classic S-shaped form.

**Activation functions** play a fundamental role in neural network training, by affecting gradient flow, expressivity, and generalization. ReLU (Nair & Hinton, 2010), despite its broad use, suffers from the well-known "dying neuron" problem, which can accelerate plasticity loss over time (Dohare et al., 2024; Sokar et al., 2023). Variants like Leaky ReLU (Maas et al., 2013), GELU (Hendrycks & Gimpel, 2016), and SiLU (a.k.a. Swish) (Elfwing et al., 2018) have sought to address these limitations. Smoother functions such as ELU (Clevert, 2015) and Mish (Misra, 2019) further enhance gradient flow and may offer better long-term learning dynamics. In contrast, classic activation functions like Sigmoid (Rumelhart et al., 1986) and Tanh (LeCun et al., 2002) have largely fallen out of favor due to issues like vanishing gradients. Yet, their behavior in continual learning remains poorly understood. It is unclear whether their saturation properties amplify plasticity loss or whether their bounded nature might offer unexpected advantages under certain conditions. While numerous previous publications (Ramachandran et al., 2017; Shang et al., 2016; Apicella et al., 2021) have benchmarked a lot of activation functions for supervised learning on static datasets, it still remains unclear how they behave in a continual or non-stationary setting. In this work, we systematically evaluate

26 activation functions under non-stationary conditions, with a particular focus on their interactions with normalization.

**Normalization layers** have become very important for modern deep learning due to their ability to stabilize training and accelerate convergence. Batch Normalization (BatchNorm) (Ioffe, 2015), despite its success in supervised learning, has been traditionally considered incompatible with reinforcement learning and continual learning due to its reliance on batch-level statistics (Ioffe, 2017; Santurkar et al., 2018). Recent results, however, challenge this view: (Bhatt et al., 2024) show that BatchNorm can be adapted to non-i.i.d. RL settings with proper handling. Conversely, Layer Normalization (LayerNorm) (Ba et al., 2016) has seen broader adoption in RL and continual learning due to its robustness to distributional shifts (Kapturowski et al., 2022; Lee et al., 2024a; Lyle et al., 2024b). Yet, the interaction between normalization schemes and plasticity loss remains underexplored. Are some activations inherently better suited for BatchNorm or LayerNorm when distribution shifts are expected? Could there be certain combinations that slow down the loss of plasticity better than others?

## 3   Experimental Setup

This section describes the experimental setup to evaluate the different model architectures. We introduce the benchmark environment, define the evaluation of plasticity loss, outline the model and training protocols, and report the computational resources used.

**Class-incremental CIFAR-100.**   Our experiments are conducted on the class-incremental variant of CIFAR-100 (Krizhevsky et al., 2009), first proposed by Dohare et al. (2024). This benchmark is explicitly suited to evaluate models under both input and output distribution shifts, which are central to the study of plasticity. In our case this refers to changes in the image and label distribution. As illustrated in Figure 2, the dataset is partitioned into 20 sequential tasks, each introducing five new classes. These staged distribution shifts simulate a realistic continual learning scenario. Unseen inputs emerge over time, and the classifier output space gradually expands as more output neurons become active. The particular class expansion order is given via seed, and starting with the last task (the 20th task), the model is training on the entire CIFAR-100 dataset. Note, that this random class expansion order introduces high variance in test accuracy at the first tasks, because certain dataset subsets are easier to classify than others. This variance shrinks significantly when the final task is reached (e.g. see Figure 1, panels on the right).

**Plasticity Evaluation.**   To quantify plasticity loss, we adopt the framework introduced by Lyle et al. (2024b). Hereby, plasticity loss is defined as a neural network's diminished ability to optimize its objective function as effectively as a randomly initialized network with the same architecture. In practical terms, we compare two training modes which are also visualized in Figure 2:

- *Reset*: a fresh, randomly initialized model is trained from scratch only on the current task.

- *No-Reset*: the same model architecture continues training sequentially across all tasks without re-initializing weights.

By comparing the test performance of both training modes on each task (see Figure 2), we measure the decline in learning ability. To ensure statistical significance, all experiments are repeated with five different seeds and we take the average test accuracy and report variance in our figures.

**Training Protocol.**   Following best practices from Dohare et al. (2024), we use several training strategies known to improve the general performance and reduce loss of plasticity. This includes standard image augmentations (random cropping, horizontal flipping, and rotation), as well as L2 regularization, which has been shown (Lyle et al., 2024b) to improve plasticity by keeping weight norms low. To ensure fair comparison across all architectures, we search over the L2 regularization parameter ($\lambda$) individually for each activation-normalization pair (see Table 7). Additionally, we tune the learning rate and schedule for each normalization setting. All models are trained for 200 epochs per task to ensure proper convergence. More details are given in Appendix B and F.

**Model Architecture.** We use the same ResNet-18 architecture as Dohare et al. (2024). The detailed layer configuration is given in Appendix C.

**Compute Resources.** All experiments were conducted on NVIDIA A100 GPUs. The full study including hyperparameter tuning required approximately 75 A100 GPU-days. A single No-Reset run takes roughly 160 minutes.

## 4 Comparison of Common and Uncommon Activation Functions

A core objective of this study is to see how different activation functions influence the network's plasticity. To this end, we start by evaluating twelve activation functions available in PyTorch, spanning both widely-used and less common variants. These functions are visualized in Figure 3. Our selection includes popular choices such as ReLU, Leaky ReLU, GELU, and SiLU, which are essential in modern deep learning architectures. In addition, we investigate several less frequently used functions like ELU, SELU, Tanh, HardTanh, Sigmoid, LogSigmoid, HardSigmoid, and Softplus that may show different characteristics. All activation functions are put into the same ResNet-18 architecture and trained across 20 tasks. For each task (see Figure 2), we compare the two training modes: *Reset* (where the model is reinitialized between tasks) and *No-Reset* (where learning continues without re-initialization). This allows us to evaluate the impact of activation choice on plasticity.

**Paper Structure.** The rest of the paper is organized as follows: Section 4.1 presents a comparative analysis of their performance under both training modes. In Section 4.2, we highlight empirical findings. After that, we dig deeper into the interesting parts of our findings and conduct additional experiments in Section 4.3, 4.4 & 4.5. Section 5 explores systematic modifications to existing functions aimed at creating novel variants to enhance plasticity. In Section 6, we introduce novel activation functions designed to improve and study performance in continual learning scenarios. Finally, Section 7 presents the results for architectures that suffered plasticity loss in previous sections. Hereby, small training modifications are added to inspect further plasticity behavior. We conclude with recommendations for model training in continual learning (see Section 8).

### 4.1 Results

This section presents the performance of the twelve activation functions introduced in Figure 3. A detailed discussion of the observed trends and related hypotheses is provided in Section 4.2. Table 1 summarizes the results for all normalization settings, respectively. In each case, we report the top-1 classification accuracy on the complete CIFAR-100 test set following the final task (20). For the *Reset* and *No-Reset* setting, we also provide a complete task-wise comparison over all 20 tasks, which can be found in Appendix D. The corresponding figures are Figure 10 for BatchNorm, Figure 12 for LayerNorm, and Figure 14 for no normalization. The key findings are summarized below:

**BatchNorm.** At the top, Table 1 reports the performance of networks trained with Batch Normalization. Under the *Reset* condition, the most widely adopted activation functions (ReLU, GELU, Leaky ReLU, SiLU, and ELU) achieve the highest top-1 accuracy, ranging from 76.71% to 74.90%. These functions are preferred for their effectiveness due to properties like non-saturation, piece-wise linearity, and mitigation of the vanishing gradient problem (He et al., 2015; Maas et al., 2013). In contrast, less commonly used functions such as Sigmoid, Softplus, etc. show higher error rates, likely due to their saturation behavior, which is known to kill gradient flow and training efficiency (LeCun et al., 2002). A detailed breakdown of performance across all activation functions under both training paradigms can be found in Figure 10.

In the *No-Reset* setting, several changes occur:

- The best-performing activations under *Reset* generally suffer a decline in performance.

| Reset (BatchNorm) | | | | | No-Reset (BatchNorm) | | | |
| --- | --- | --- | --- | --- | --- | --- | --- | --- |
| **Rank** | **Top-1 Acc. (%)** | **Activation** | **↑ / ↓** | | **Rank** | **Top-1 Acc. (%)** | **Activation** | **↑ / ↓** |
| 1 | 76.71 | ReLU | | | 1 | 75.66 (-0.73) | GELU | ▲2 |
| 2 | 76.48 | Leaky ReLU | | | 2 | 75.55 (-0.40) | SiLU | ▲2 |
| 3 | 76.39 | GELU | | | 3 | 75.42 (+0.52) | ELU | ▲2 |
| 4 | 75.95 | SiLU | | | 4 | 74.62 (-1.86) | Leaky ReLU | ▼2 |
| 5 | 74.90 | ELU | | | 5 | 74.30 (+1.30) | Softplus | ▲4 |
| 6 | 73.88 | Tanh | | | 6 | 74.23 (+0.35) | Tanh | 0 |
| 7 | 73.63 | SELU | | | 7 | 74.17 (+1.67) | LogSigmoid | ▲3 |
| 8 | 73.42 | HardTanh | | | 8 | 73.92 (+4.15) | Sigmoid | ▲3 |
| 9 | 73.00 | Softplus | | | 9 | 73.60 (-0.03) | SELU | ▼2 |
| 10 | 72.50 | LogSigmoid | | | 10 | 73.42 (+0.00) | HardTanh | ▼2 |
| 11 | 69.77 | Sigmoid | | | 11 | 72.06 (-4.65) | ReLU | ▼10 |
| 12 | 64.00 | HardSigmoid | | | 12 | 71.50 (+7.50) | HardSigmoid | 0 |
| **Reset (LayerNorm)** | | | | | **No-Reset (LayerNorm)** | | | |
| 1 | 73.70 | Leaky ReLU | | | 1 | 73.21 (+0.02) | GELU | ▲2 |
| 2 | 73.48 | ReLU | | | 2 | 71.86 (-1.27) | SiLU | ▲2 |
| 3 | 73.19 | GELU | | | 3 | 70.34 (-2.02) | ELU | ▲2 |
| 4 | 73.13 | SiLU | | | 4 | 67.40 (-2.97) | SELU | ▲2 |
| 5 | 72.36 | ELU | | | 5 | 67.14 (+1.42) | Tanh | ▲2 |
| 6 | 70.37 | SELU | | | 6 | 63.24 (-1.09) | HardTanh | ▲2 |
| 7 | 65.72 | Tanh | | | 7 | 62.08 (-0.04) | LogSigmoid | ▲3 |
| 8 | 64.33 | HardTanh | | | 8 | 62.01 (-0.78) | Softplus | ▲1 |
| 9 | 62.79 | Softplus | | | 9 | 10.45 (-63.25) | Leaky ReLU | ▼8 |
| 10 | 62.12 | LogSigmoid | | | 10 | 01.00 (-72.48) | ReLU | ▼8 |
| **Reset (No-Norm)** | | | | | **No-Reset (No-Norm)** | | | |
| 1 | 68.68 | SELU | | | 1 | 64.71 (-3.97) | SELU | 0 |
| 2 | 66.72 | ELU | | | 2 | 63.21 (+0.12) | HardTanh | ▲1 |
| 3 | 63.09 | HardTanh | | | 3 | 62.59 (+0.32) | Tanh | ▲1 |
| 4 | 62.91 | Tanh | | | 4 | 62.08 (-4.64) | ELU | ▼2 |
| 5 | 62.63 | ReLU | | | 5 | 60.45 (-2.18) | ReLU | 0 |
| 6 | 62.51 | Leaky ReLU | | | 6 | 60.03 (-2.48) | Leaky ReLU | 0 |
| 7 | 59.38 | GELU | | | 7 | 55.35 (-4.03) | GELU | 0 |
| 8 | 58.76 | SiLU | | | 8 | 53.54 (-5.22) | SiLU | 0 |

Table 1: **BatchNorm, LayerNorm & No-Norm**. This table reports the Top-1 accuracy (%) of ResNet-18 on the CIFAR-100 test set after task 20. Each value represents the mean accuracy over five seeds. The column labeled "↑/↓" indicates changes in relative rank (*Reset* vs. *No-Reset*), with symbols denoting the direction of change: ▲ for improvement, ▼ for decrease, and zero for no change. The accuracy coloring for *No-Reset* runs represents the plasticity behavior: (1) Fast Decline, (2) Slow Decline, (3) Slight Improvement or On-Par, and (4) Improvement. Activation functions are omitted when unstable or failed training occurred for a particular configuration.

- Interestingly, some of the weaker activations under *Reset* (ELU, Tanh, LogSigmoid, Softplus, and Sigmoid) show improved plasticity. They benefit from the training on previous tasks, improving their test accuracy by up to 4% in top-1 error compared to their *Reset* counterparts.

- GELU rises in ranking, becoming the top performer in the *No-Reset* condition with 75.66% in top-1 accuracy. Closely followed by SiLU and ELU.

- ReLU experiences a significant drop, falling ten positions in rank. Its accuracy decreases from 76.71% to 72.08% which is a decline of approximately 4.65%.

- Leaky ReLU also drops in absolute performance by 1.86%.

**LayerNorm.** We evaluate the effectiveness of Layer Normalization as an alternative to Batch Normalization, with results presented in the middle section of Table 1. Models using LayerNorm generally have lower top-1 accuracy on this benchmark compared to those using BatchNorm, confirming previous results (Liu et al.,

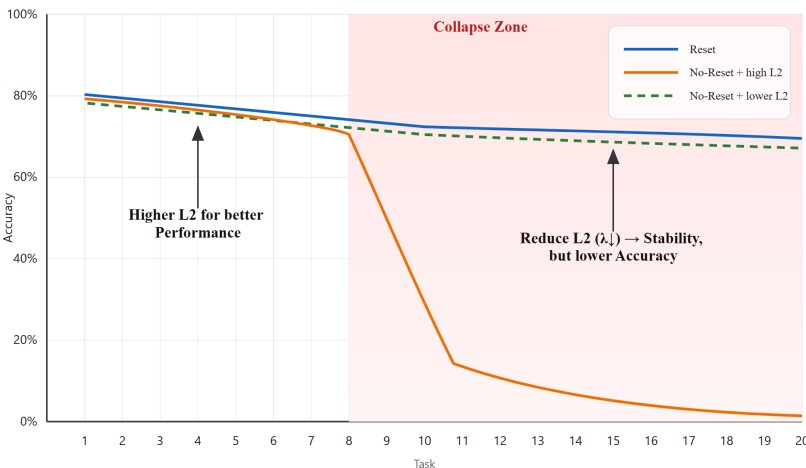

Figure 4: **Schematic of L2 Collapse in Continual Learning (with LayerNorm).** ResNet-18 trained on class-incremental CIFAR-100 under No-Reset sometimes suffers an abrupt accuracy drop (characterized as "Collapse Zone") when L2 regularization is tuned for full-dataset performance and paired with ReLU or Leaky ReLU. The plot compares Reset (blue), No-Reset + high L2 (orange, collapses), and No-Reset + lower L2 (green, training remains stable). The shaded region marks the region where multiple runs consistently fail, highlighting the stability-performance trade-off. Reducing L2 regularization prevents collapse yet sacrifices accuracy. Please, see Figure 16 and 17 for the actual data.

2022). While Batch Normalization performs well in standard classification, it is often unsuitable for settings with online updates, small batches, or non-stationary data like in Reinforcement Learning. In such cases, Layer Normalization is typically favored for its batch-independent statistics (Kapturowski et al., 2022; Lee et al., 2024a; Lyle et al., 2024b). In the *Reset* setting, the relative performance of activation functions largely mirrors the trends observed with BatchNorm. GELU, ReLU, Leaky ReLU, and SiLU achieve the best results with top-1 accuracy ranging from 73.70% to 73.13%. Other activation functions perform noticeably worse. Figure 12 provides a detailed comparison between the *Reset* and *No-Reset* settings.

In the *No-Reset* setting, several changes occur:

- Models using ReLU or Leaky ReLU experience training instability that ultimately leads to collapse. We explore this phenomenon in Section 4.3 & 4.4 and sketch it in Figure 4.

- GELU keeps its strong performance, achieving a 73.21% top-1 accuracy. This is almost identical to its result in the *Reset* scenario (73.19%).

- A subset of activations (SiLU, ELU, SELU, Softplus, and HardTanh) show a notable degradation, indicating a loss of plasticity relative to their *Reset* baseline.

- In contrast, Tanh and LogSigmoid keep stable performance across both training regimes.

**No-Norm.** When evaluating models without any normalization layers (see Table 1), we observe a notable shift in activation function performance. In the *Reset* setting, SELU achieves the highest top-1 accuracy (68.68%) that outperforms commonly dominant functions like ReLU and GELU. This highlights the extent to which these popular activations rely on normalization to perform effectively.

Under the *No-Reset* setting, several changes occur:

- SELU stays at the top position but suffers a noticeable performance drop (accuracy decreases to 65.10%), suggesting that its self-normalizing behavior is insufficient to fully preserve plasticity across tasks.

- ReLU, Leaky ReLU, GELU, and SiLU also experience significant error increases, highlighting their dependence on normalization layers.

- In contrast, HardTanh and Tanh show greater robustness and maintain their performance even in the absence of normalization layers.

## 4.2 Observations

In this section, we present a set of preliminary observations derived from the experimental results. While we attempt to interpret these findings carefully, we emphasize that some of the trends are subtle, and the complexity of these results does not always allow a clear conclusion. These insights nevertheless have a useful role: they help us to formulate new hypotheses and guide the design of more targeted experiments in the following sections.

**General observation.** We consistently observe a noticeable correlation between a model's ability to minimize the cross-entropy loss and a subsequent loss in plasticity (see Figure 11, 13, & 15). Specifically, architectures that achieve very low training loss tend to suffer from reduced plasticity. This effect is happening across all evaluated scenarios, including BatchNorm, LayerNorm, and without normalization. However, the loss threshold at which plasticity degradation occurs varies depending on the employed normalization technique. These results indicate that while low training loss is desirable from an optimization perspective, it could reduce the model's adaptability in continual learning.

**BatchNorm.** We observe a distinction in behavior between activation functions based on the presence or absence of the origin property (see Appendix A). Specifically, activation functions that do not exhibit the origin property tend to remain plastic across all sequential tasks. Among these, we identify two activation functions (Softplus, LogSigmoid) that not only lack the origin property but also achieve a low training loss while still preserving plasticity (see Figure 11). Conversely, activation functions that have both a very low training loss and the origin property demonstrate a reduction in plasticity. Interestingly, this effect disappears for activation functions with the origin property if their training loss is comparatively higher.

**LayerNorm.** The behavior observed under Layer Normalization is less consistent and more difficult to interpret. In particular, we note that the use of L2 regularization (see Section 4.3) appears to introduce serious instability for certain activation functions, such as ReLU and LeakyReLU (see Table 1 & Figure 12). Although prior work shows that LayerNorm indeed helps to preserve plasticity in continual learning settings (Lyle et al., 2024b), our results indicate that this beneficial effect is sensitive to hyperparameter choices and can be detrimental when chosen wrong.

**No-Norm.** Activation functions such as GELU and SiLU show a strong reliance on normalization layers for both optimal performance and sustained plasticity. In the absence of normalization, we observe a consistent degradation in performance, with absolute errors increasing by 4–5%. This indicates a significant reduction in the model's ability to adapt and generalize. While SELU remains the most stable activation in this setting, likely due to its self-normalizing properties (Klambauer et al., 2017), it also experiences performance degradation under distribution shifts. These findings suggest that implicit normalization alone is insufficient, and highlight the important role of explicit normalization layers in preserving network plasticity.

## 4.3 L2 Collapse

**Experimental Reminder.** Our *No-Reset* setup trains models sequentially on CIFAR-100 tasks. For each architecture, the L2 regularization coefficient is first tuned for best performance on the *full* CIFAR-100 dataset with selected values reported in Table 7.

**Training Instability.** In the continual learning setting, we observe a severe optimization failure that we call *L2 collapse.* It affects models with ReLU or Leaky ReLU activations when paired with Layer Normalization (see Figure 4). While early tasks train normally, performance sharply decreases after about nine tasks and

the models become nearly untrainable. The training loss plateaus at a high value which indicates that the optimizer fails to make meaningful progress. This occurs despite a learning rate schedule that reduces the rate by two orders of magnitude (see Figure 18), pointing to a deeper optimization issue.

**Stability vs. Performance.** To prevent *L2 collapse* and maintain trainability throughout the entire task sequence, we found that it is necessary to reduce the L2 coefficient substantially. Specifically, it needs to be reduced by a factor of at least three relative to the value optimized on the full dataset. While this stabilizes training, it comes at the cost of final performance: top-1 accuracy decreases to 71.49% (ReLU) and 71.61 % (Leaky ReLU), ranking behind GELU and SiLU. More importantly, Section 6 introduces ReLU variants that improve over these results, indicating that the reduced regularization indeed leads to sub-optimal performance (see Table 2). Overall, these results underline the need to reconsider fixed regularization strategies in continual learning. Adaptive (L2) regularization may be interesting for balancing stability and generalization in such dynamic training settings.

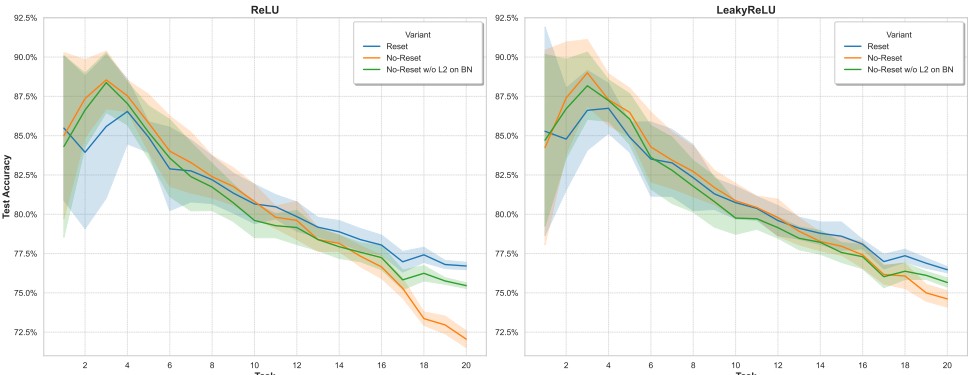

Figure 5: **BatchNorm. Excluding L2 Regularization on Affine Parameters Improves Plasticity.** The left panel shows results for networks using BatchNorm with ReLU activations; the right panel shows the same setup with LeakyReLU. Each side compares three training configurations: (1) *Reset*; (2) *No-Reset*; and (3) *No-Reset w/o L2 on BN*, identical to (2) and with the *same* L2 strengths, but excluding L2 regularization on BatchNorm's affine parameters (scale and shift). In both cases, removing L2 on BatchNorm improves plasticity compared to classic *No-Reset*, but it still underperforms the *Reset* baseline.

## 4.4 L2 Regularization and Normalization Layers

We observe (see Tables 1 & Figure 10, 12) that networks using normalization layers still suffer from a loss of plasticity, especially when paired with certain activation functions. Motivated by the hypothesis that this degradation may originate from overfitting to the current task or other adaptability issues (see Section 4.3), we explored several ideas to reduce this effect.

**Key Finding.** We found that excluding L2 regularization from the affine parameters (scale and shift) of normalization layers, both for BatchNorm and LayerNorm, consistently improves plasticity (see Figure 5 & 6). This suggests that regularizing these parameters may unnecessarily restrict the network's ability to adapt and that lifting this constraint allows for more flexible adjustment during training.

**BatchNorm.** As shown in Figure 5, removing L2 regularization from BatchNorm's affine parameters significantly improves performance under the *No-Reset* setting. This effect is consistent across two activation functions (ReLU and LeakyReLU) that show the greatest loss of plasticity. This improvement narrows the gap to the *Reset* baseline significantly, but is not able to fully overcome it.

**LayerNorm.** Similar benefits are observed when excluding L2 from LayerNorm's affine parameters (see Figure 6). This was particularly evident with activation functions like SiLU and SELU, which previously

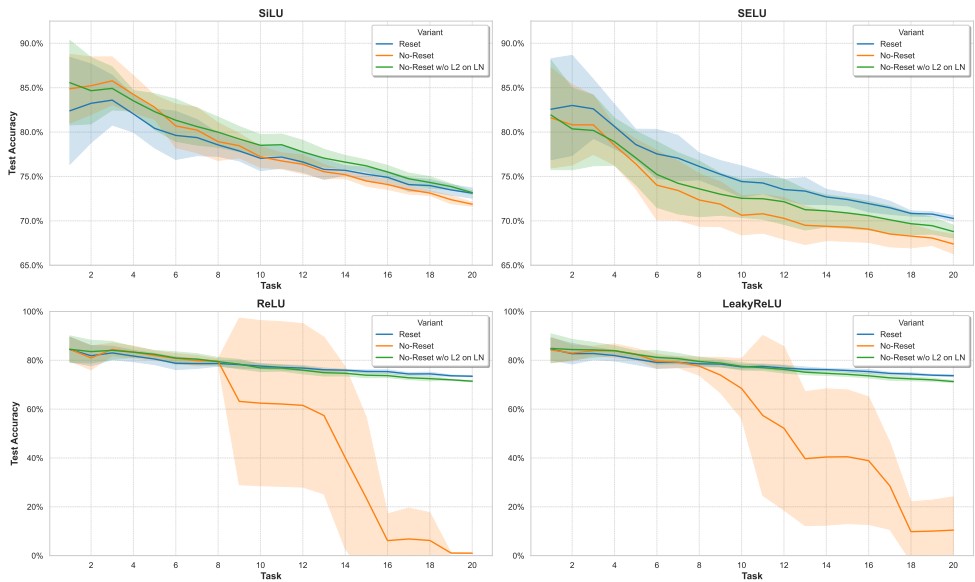

Figure 6: **LayerNorm. Excluding L2 Regularization on Affine Parameters Improves Plasticity.** This figure shows results for networks using LayerNorm across four different activation functions. Each panel compares three training configurations: (1) *Reset*; (2) *No-Reset*; and (3) *No-Reset w/o L2 on LN*, identical to (2) and with the *same* L2 strengths, but excluding L2 regularization on LayerNorm's affine parameters (scale and shift). In all cases, removing L2 on LayerNorm improves plasticity compared to classic *No-Reset*. Using SiLU, this tiny modification achieves on-par results with the *Reset* setting.

showed a noticeable loss in plasticity under the *No-Reset* training. The results indicate that SiLU reaches performance on-par with its *Reset* baseline when L2 regularization is removed from the affine parameters. SELU also benefits, showing a decrease of the performance gap. Additionally, we observe that removing L2 from LayerNorm's affine parameters helps to prevent L2 collapse (see Section 4.3), a sharp degradation in performance at higher L2 strengths. This suggests that over-regularizing LayerNorm's affine parameters can destabilize training, and that excluding them improves both adaptability and robustness.

**General Insight.** By simply excluding the affine parameters from L2, we observe improved plasticity, training stability, and more flexibility in applying stronger regularization to other parts of the network. This intervention generalizes well across normalization types and activation functions and presents a minimal yet impactful design choice.

## 4.5 On the Relationship between Plasticity and Overfitting

**Overfitting vs. Plasticity.** We observed a trend: models that minimize the training loss too aggressively tend to suffer more from reduced plasticity (see Figures 11, 13 & 15). Conversely, models that stay at a relatively high training loss often maintain or even improve their performance across tasks (as in Table 1, Figure 10, 12 & 14). This observation motivates us to explore strategies that explicitly regularize the magnitude of the training loss. To this end, we apply label smoothing (Szegedy et al., 2016) as a form of output regularization and re-evaluate plasticity in this context. Using a very good but less plastic activation function, Figure 7 shows training runs for models using ReLU activation with BatchNorm and varying degrees of label smoothing. Even small amounts of label smoothing noticeably improve performance in the continual learning setting. As smoothing ($\varepsilon$) increases, both robustness and overall performance improve, indicating that label smoothing not only acts as a regularizer but also seem to reduce the effects of degradation. Smoothing values beyond $\varepsilon = 0.2$ yield no additional performance gains but greater values could *potentially* further increase plasticity as a trade-off.

**Latent Plasticity.** However, the observed performance improvements with label smoothing do not necessarily imply that these models maintain a higher degree of plasticity compared to their non-regularized counterparts. It is known that label smoothing, in general, can slightly increase performance (Szegedy et al., 2016) for some activation functions like ReLU. To further investigate this hypothesis, we look at a non-optimal but stronger smoothing value ($\varepsilon = 0.4$), which yields lower test accuracy (see Figure 7, right) than the optimal value. To verify whether plasticity is indeed preserved in these models ($\varepsilon = 0.4$) despite their sub-optimal performance, we conduct an intervention. Almost at the end of every training run, we create the possibility to disable label smoothing, thereby allowing the model to minimize the training loss more aggressively again. In our case, we set the start of task 18 as the intervention point so that the last 400 epochs are either continued with label smoothing or it is disabled.

**Improvement.** Interestingly, this late-stage intervention results in a clear improvement in Top-1 accuracy (see Figure 7). Specifically, models trained with smoothing intervention (denoted in yellow) surpass the baseline ($\varepsilon = 0.2$) by an absolute margin of 0.5% test accuracy. Notably, such late-stage improvements have not been observed in any other training runs. This finding supports the hypothesis that label smoothing can preserve latent plasticity and it can be recovered once the regularization constraint is lifted. If label smoothing had not contributed to the preservation of plasticity, no performance gain would be expected following its removal. Especially, since models trained without any smoothing perform a lot worse in general (see Figure 7, left). These results reveal a trade-off between immediate performance and long-term adaptability. Output regularization via label smoothing can serve as an effective way to delay overfitting and increase plasticity over extended task sequences.

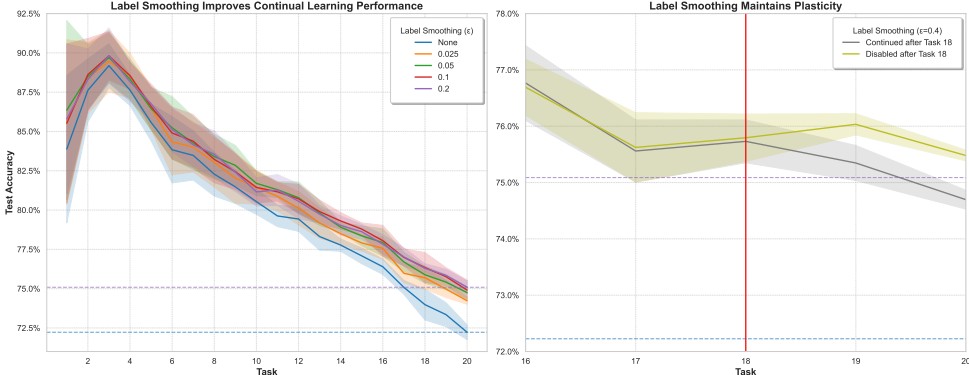

Figure 7: **Label Smoothing Improves Both Performance and Plasticity.** We evaluate ResNet-18 with ReLU and BatchNorm in the *No-Reset* setting. The left panel compares test accuracy for different label smoothing strengths. The right panel zooms in on the final five tasks to evaluate long-term adaptability with an intervention. After task 18, models either continue to train with label smoothing or disable it ($\varepsilon \to 0$). Disabling label smoothing results in improved performance, beating the previously best-performing value ($\varepsilon = 0.2$). Horizontal dotted lines highlight the accuracy gain from using label smoothing over no smoothing.

## 5 Modding Activation Functions

While standard activation functions like ReLU or GELU remain widely used due to their empirical success and somewhat theoretical grounding (Glorot et al., 2011; He et al., 2015; Lee, 2023; Dubey et al., 2022), exploring modifications to these functions may offer new insights into plasticity behavior and failure modes in continual learning. In this section, we introduce a set of simple but expressive transformation types referred to as *activation mods*. These allow us to systematically alter existing activation functions. The modifications serve two purposes: (1) identifying characteristics that reduce or enhance plasticity, and (2) enabling the design of potentially more adaptable activation functions without sacrificing performance.

### 5.1 Mod Types

Let $\phi : \mathbb{R} \to \mathbb{R}, x \mapsto \phi(x)$ denote a generic activation function, which maps real-valued inputs to real-valued outputs. The following modifications offer a lightweight yet flexible framework to manipulate and potentially enhance activation functions for continual learning.

**Shift.** This mod changes the input and output bias of an activation function:

$$S(\phi, \alpha, \beta)(x) := \phi(x + \alpha) + \beta, \tag{1}$$

where $\alpha \in \mathbb{R}$ shifts the input domain and $\beta \in \mathbb{R}$ adjusts the output baseline. This can affect symmetry and saturation properties, both of which may influence plasticity.

**Linearize.** This mod approximates a nonlinear activation by a piece-wise linear function:

$$L(\phi)(x) := \begin{cases} a_1 x + b_1, & x \in (-\infty, x_1], \\ a_2 x + b_2, & x \in [x_1, x_2], \\ \vdots \\ a_n x + b_n, & x \in [x_{n-1}, \infty). \end{cases} \tag{2}$$

The segmentation and coefficients $\{a_i, b_i\} \in \mathbb{R}$ are chosen to approximate $\Phi$ under a specified criterion (e.g., least squares or personal preference). This mod allows for more interpretable activations and facilitates analytical comparisons.

**Compose.** This mod creates a new activation function by combining two existing ones using a composition operator $g : (\mathbb{R} \to \mathbb{R}) \times (\mathbb{R} \to \mathbb{R}) \to \mathbb{R}$:

$$C(\phi_1, \phi_2, g)(x) := g(\phi_1, \phi_2)(x), \tag{3}$$

For example, $g$ could be addition $(\phi_1(x) + \phi_2(x))$, multiplication $(\phi_1(x) \cdot \phi_2(x))$, or maximum $(\max(\phi_1(x), \phi_2(x)))$. Function chaining (i.e., $\phi_2(\phi_1(x))$) is also a special case of composition.

**Drift.** The drift mod adds a secondary signal to the activation output:

$$D(\phi, f, \alpha)(x) := \phi(x) + \alpha f(x) \tag{4}$$

where $f : \mathbb{R} \to \mathbb{R}$ is a drift function, and $\alpha \in \mathbb{R}$ controls its influence. The drift term allows dynamic adjustment of activation behavior and can be seen as a generalization of residual or bias-like effects.

## 6  Studying Plasticity with New Activation Functions

The proposed modifications allow a systematic exploration of how subtle changes to activation functions affect both performance and plasticity. To illustrate this, we present mods for new activations that are shown in two figures: Figure 8 focuses on ReLU and its variants, while Figure 9 highlights modifications to GELU along with several other newly designed functions. These new variants are build to test underexplored areas such as asymmetry, smoothness, changes in asymptotic behavior, or non-zero baselines. Each following subsection introduces the rationale behind certain modifications and presents comparative empirical results. It then concludes with short takeaways regarding its impact on plasticity and performance. Later in Section 7, we report additional modified training runs specifically for plasticity affected activation functions.

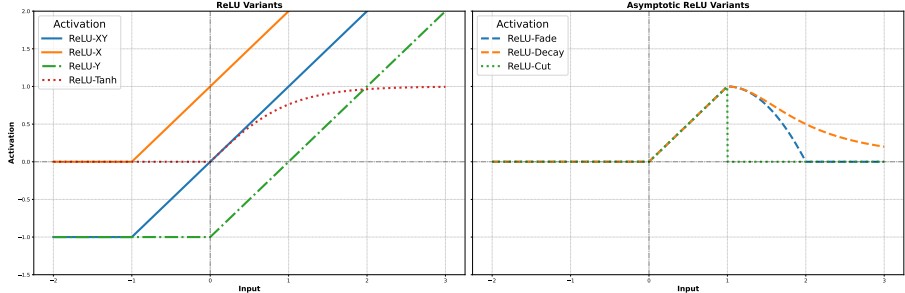

Figure 8: **Modified ReLU Variants.** The left panel illustrates several *Shifted ReLU* modifications, including horizontal shifts, vertical offsets, and the *Tanh-ReLU* hybrid. The right panel shows *Asymptotic ReLU* variants, where the linear growth of ReLU is truncated beyond a threshold (e.g., $x \geq 1$), causing the activation to saturate and move down to zero.

## 6.1  Modding ReLU

**Motivation.**  ReLU remains a foundational activation function in deep learning due to its simplicity and efficiency. However, in continual learning settings (*No-Reset*) it often shows poor adaptability even after excluding affine parameters from L2 regularization. This raises at least two questions:

1. Is ReLU's hinge at zero inherently bad for plasticity?

2. Does its unbounded positive linearity contribute to this problem?

To investigate these questions, we introduce several modified versions of ReLU (shown in Figure 8) that selectively modify either the hinge location or the slope of the linear region. Our goal is not only to recover plasticity but to identify minimal yet effective design changes that reduce ReLU's failure modes in continual learning. The shift modifications used are formally defined as follows:

---
**ReLU: Shift Variants**

$$ReLU\text{-}X(x) := \max(0, x + 1) \tag{5}$$
$$ReLU\text{-}Y(x) := \max(0, x) - 1 \tag{6}$$
$$ReLU\text{-}XY(x) := \max(-1, x) \tag{7}$$

---

In addition, we explore how ReLU's unbounded linear growth on the positive side affects plasticity. To this end, we design several modified versions that replace the standard linearity with saturating behaviors that asymptotically approach a finite value or zero. For some functions, this "stopping" mechanism is controlled with a threshold parameter $\tau > 0$, beyond which the activation begins to flatten or decay. We set $\tau = 3$ across all variants based on preliminary sweeps. Smaller values ($\tau < 2$) often led to poor training results, likely due to conflicts with normalization layers or weight initialization. Below, we define the set of ReLU modifications:

---

**ReLU: Asymptote Variants**

$$ReLU\text{-}Tanh(x) := \max(0, \tanh(x)) \tag{8}$$

$$ReLU\text{-}Cut(x; \tau) := \begin{cases} \mathrm{ReLU}(x), & x \le \tau \\ 0, & x > \tau \end{cases} \tag{9}$$

$$ReLU\text{-}Fade(x; \tau) := \begin{cases} \mathrm{ReLU}(x), & x \le \tau \\ \tau \cdot \left(1 - (x - \tau)^2\right), & \tau < x \le \tau + 1 \\ 0, & x > \tau + 1 \end{cases} \tag{10}$$

$$ReLU\text{-}Decay(x; \tau) := \begin{cases} \mathrm{ReLU}(x), & x \le \tau \\ \frac{\tau}{1 + (x - \tau)^2}, & x > \tau \end{cases} \tag{11}$$

---

| | Reset (BatchNorm) | | | | No-Reset (BatchNorm) | | |
|---|---|---|---|---|---|---|---|
| **Rank** | **Top-1 Acc. (%)** | **Activation** | ↑ / ↓ | **Rank** | **Top-1 Acc. (%)** | **Activation** | ↑ / ↓ |
| 1 | 76.71 | ReLU | | 1 | 75.69 (-0.47) | ReLU-Cut | ▲2 |
| 2 | 76.26 | ReLU-Decay | | 2 | 75.17 (+0.03) | ReLU-Y | ▲4 |
| 3 | 76.16 | ReLU-Cut | | 3 | 74.73 (+0.53) | ReLU-XY | ▲4 |
| 4 | 75.85 | ReLU-Tanh | | 4 | 72.06 (-4.65) | ReLU | ▼3 |
| 5 | 75.64 | ReLU-Fade | | 5 | 70.48 (+1.84) | ReLU-X | ▲3 |
| 6 | 75.20 | ReLU-Y | | 6 | 70.47 (-5.17) | ReLU-Fade | ▼1 |
| 7 | 74.20 | ReLU-XY | | 7 | 70.46 (-5.80) | ReLU-Decay | ▼5 |
| 8 | 68.64 | ReLU-X | | 8 | 66.13 (-9.72) | ReLU-Tanh | ▼4 |
| | **Reset (LayerNorm)** | | | | **No-Reset (LayerNorm)** | | |
| 1 | 73.48 | ReLU | | 1 | 73.47 (+1.01) | ReLU-Tanh | ▲2 |
| 2 | 72.75 | ReLU-Cut | | 2 | 73.03 (+0.28) | ReLU-Cut | 0 |
| 3 | 72.46 | ReLU-Tanh | | 3 | 72.65 (+0.65) | ReLU-Decay | ▲1 |
| 4 | 72.00 | ReLU-Decay | | 4 | 71.65 (+0.05) | ReLU-Fade | ▲1 |
| 5 | 71.60 | ReLU-Fade | | 5 | 68.92 (2.67) | ReLU-XY | ▲1 |
| 6 | 71.59 | ReLU-XY | | 6 | 67.36 (1.50) | ReLU-Y | ▲1 |
| 7 | 68.86 | ReLU-Y | | 7 | 63.10 (+0.49) | ReLU-X | ▲1 |
| 8 | 62.61 | ReLU-X | | 8 | 1.06 (-72.42) | ReLU | ▼7 |

Table 2: **BatchNorm & LayerNorm**. We apply the same setup as in Table 1 to evaluate the new ReLU variants in terms of performance and plasticity loss.

**Results.** The effectiveness of ReLU modifications varies considerably depending on the normalization strategy. Under BatchNorm (Table 2), the original ReLU remains the strongest performer in the *Reset* condition. All modified variants underperform in this static setting, suggesting that the standard ReLU configuration is well-optimized for stable and single-task training. However, this situation changes under the more challenging *No-Reset* setting. Here, both the original ReLU and its Tanh-truncated variant (ReLU-Tanh) experience a substantial drop in plasticity due to L2 regularization issues as discussed earlier. In contrast, variants such as ReLU-X, ReLU-XY, and ReLU-Y remain plastic and demonstrate significantly more robust behavior across sequential tasks. Notably, ReLU-Cut achieves performance on-par with GELU (see Table 1) and is also slightly better than ReLU + excluding affine from L2 regularization (Figure 5). With LayerNorm, the dynamics become more nuanced (Table 2). Surprisingly, ReLU-Tanh becomes the best option under *No-Reset*, slightly improving over GELU and matching the performance of ReLU under full resets (see Table 1).

Taken together, these results suggest that while ReLU remains a strong baseline in stable environments, its characteristics under distributional shifts limit its use in continual learning. Variants such as ReLU-Cut,

ReLU-Y, and ReLU-Tanh may offer more compelling alternatives. Rather than searching for a universally optimal activation function, our findings point to the importance of aligning activation design to the specific architecture.

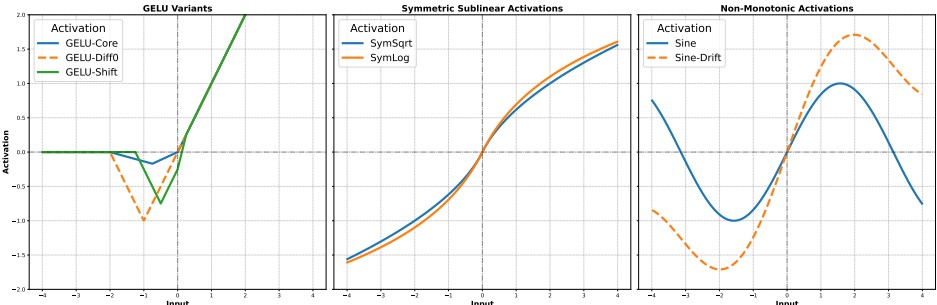

Figure 9: **Modified GELU Variants and Alternative Activation Functions.** The left panel shows novel piecewise-linear approximations of the GELU function, including *GELU-Core*, which is designed to closely match the original shape. The central panel highlights symmetric, sublinear activations, SymSqrt and SymLog, which reduce large-magnitude responses. The right panel features non-monotonic activation functions that introduce oscillatory behavior, potentially enhancing implicit regularization.

## 6.2  Modding GELU

**Motivation.**  GELU has consistently demonstrated strong empirical performance in combination with both BatchNorm and LayerNorm (see Table 1). In contrast to ReLU, which is non-differentiable at $x = 0$, GELU is continuously differentiable everywhere. We hypothesize that its smoothness may contribute to GELU's superiority in *No-Reset* settings. To investigate this, we construct three piecewise-linear versions of GELU (shown in Figure 9) that vary in their differentiability and placement of non-smooth points. These variants allow us to isolate which aspects of GELU's behavior could be most important:

- *GELU-Core*: A close linear approximation of GELU's overall shape.

- *GELU-Diff0*: A variant designed to be smooth at $x = 0$, testing whether differentiability at the origin is important.

- *GELU-Offset*: Builds on GELU-Diff0 but shifts the central region away from zero and does not pass the origin.

---

**GELU: Linearity Variants**

$$GELU\text{-}Core(x) := \begin{cases} 0 & \text{if } x < -2.0, \\ -0.136\,x - 0.272 & \text{if } -2.0 \le x < -0.75, \\ 0.227\,x & \text{if } -0.75 \le x < 0.0, \\ x & \text{if } x \ge 0.0. \end{cases} \tag{12}$$

$$GELU\text{-}Diff0(x) := \begin{cases} \min(0, |x + 1| - 1) & \text{if } x < 0, \\ \max(0, x) & \text{if } x \ge 0. \end{cases} \tag{13}$$

$$GELU\text{-}Shift(x) := \begin{cases} \min(0, |x + 0.5| - 0.75) & \text{if } x < 0, \\ \max(0, x) & \text{if } x \ge 0. \end{cases} \tag{14}$$

---

| Reset (BatchNorm) | | | | No-Reset (BatchNorm) | | | |
|---|---|---|---|---|---|---|---|
| Rank | Top-1 Acc. (%) | Activation | ↑ / ↓ | Rank | Top-1 Acc. (%) | Activation | ↑ / ↓ |
| 1 | 76.39 | GELU | | 1 | **75.83** (+0.15) | GELU-Core | ▲₁ |
| 2 | 75.98 | GELU-Core | | 2 | **75.66** (-0.73) | GELU | ▼₁ |
| 3 | 72.43 | GELU-Diff0 | | 3 | **73.39** (+1.04) | GELU-Shift | ▲₁ |
| 4 | 72.35 | GELU-Shift | | 4 | **73.11** (+0.68) | GELU-Diff0 | ▼₁ |
| Reset (LayerNorm) | | | | No-Reset (LayerNorm) | | | |
| 1 | 73.19 | GELU | | 1 | **73.21** (+0.02) | GELU | 0 |
| 2 | 72.46 | GELU-Core | | 2 | **72.92** (+0.46) | GELU-Core | 0 |
| 3 | 69.08 | GELU-Diff0 | | 3 | **68.16** (-0.42) | GELU-Shift | ▲₁ |
| 4 | 68.58 | GELU-Shift | | 4 | **64.56** (-4.52) | GELU-Diff0 | ▼₁ |

Table 3: **BatchNorm & LayerNorm**. We apply the same setup as in Table 1 to evaluate the new GELU variants in terms of performance and plasticity loss.

**Results.** To test the importance of continuous differentiability, we introduced several piecewise-linear GELU variants (see Figure 9). Surprisingly, the results show that smoothness is not a critical factor. In particular, GELU-Core consistently matches GELU's performance across both BatchNorm and LayerNorm runs (see Table 3). This suggests that GELU's effectiveness stems more from its overall shape and the resulting gradient behavior than from its mathematical smoothness. However, approximations that deviate more the original shape (GELU-Diff0 and GELU-Shift) do show performance drops for LayerNorm while almost sharing the same L2 regularization coefficient w.r.t. GELU and GELU-Core (see Table 7).

## 6.3 Other Activation Functions

**Motivation.** To conclude our study, we explore a few less conventional activation functions that introduce qualitatively different characteristics. In particular, we consider *periodic* and *sublinear-divergent* functions. These classes are motivated by the idea that certain mathematical properties, like repeating patterns or gradual divergence, might improve robustness to distribution shifts or influence plasticity in unique ways.

**Periodicity.** Periodic functions, by nature, revisit prior output values. This cyclic behavior could potentially counteract the effects of distributional drift in continual learning. To evaluate this, we define two periodic activations and analyze their performance in our setup:

> **Periodicity Variants**
>
> $$Sine(x) := \sin(x) \tag{15}$$
> $$Sine\text{-}Drift(x) := \sin(x) + x \tag{16}$$

**Sublinearity and Divergence.** We also explore functions that grow sublinearly but diverge over time. These may offer a middle ground between saturating and unbounded activations. Two such examples are given below with smooth and symmetric characteristics. Here, we include the *SymLog* function originally introduced in DreamerV3 (Hafner et al., 2023) for reward scaling:

> **Sublinear and Divergent Variants**
>
> $$SymSqrt(x) := \text{sign}(x) \left( \sqrt{|x| + a} - \sqrt{a} \right) \tag{17}$$
> $$SymLog(x) := \text{sign}(x) \log (|x| + 1) \tag{18}$$

| Reset (BatchNorm) | | | | | No-Reset (BatchNorm) | | | |
|---|---|---|---|---|---|---|---|---|
| Rank | Top-1 Acc. (%) | Activation | ↑ / ↓ | | Rank | Top-1 Acc. (%) | Activation | ↑ / ↓ |
| 1 | 76.71 | ReLU | | | 1 | 75.66 (-0.73) | GELU | ▲₁ |
| 2 | 76.39 | GELU | | | 2 | 73.02 (+0.45) | Sine-Drift | ▲₁ |
| 3 | 72.57 | Sine-Drift | | | 3 | 72.97 (+1.27) | SymLog | ▲₁ |
| 4 | 71.70 | SymLog | | | 4 | 72.50 (+1.05) | Sine | ▲₁ |
| 5 | 71.45 | Sine | | | 5 | 72.06 (-4.65) | ReLU | ▼₄ |
| 6 | 70.22 | SymSqrt | | | 6 | 71.68 (+1.46) | SymSqrt | ₀ |
| Reset (LayerNorm) | | | | | No-Reset (LayerNorm) | | | |
| 1 | 73.48 | ReLU | | | 1 | 73.21 (+0.02) | GELU | ▲₁ |
| 2 | 73.19 | GELU | | | 2 | 67.82 (+0.68) | SymSqrt | ▲₂ |
| 3 | 67.48 | SymLog | | | 3 | 67.40 (-0.08) | SymLog | ₀ |
| 4 | 67.14 | SymSqrt | | | 4 | 66.50 (-0.18) | Sine | ▲₁ |
| 5 | 66.68 | Sine | | | 5 | 58.35 (-6.46) | Sine-Drift | ▲₁ |
| 6 | 64.81 | Sine-Drift | | | 6 | 1.06 (-72.42) | ReLU | ▼₅ |

Table 4: **BatchNorm & LayerNorm**. We apply the same setup as in Table 1 to evaluate the the other experimental variants in terms of performance and plasticity loss.

**Results.** Periodic and sublinear-divergent activation functions show an interesting property: they generally do not show a degree of plasticity loss (see Table 4) as commonly seen in standard activations (see Table 1). Merely the combination Sine-Drift with LayerNorm experiences loss of plasticity, but it has a lower cross entropy loss compared to the others (see Figure 13). However, despite this benefit, these activations fall short in terms of overall performance. Models using them typically show higher training loss (see Figure 11 & 13) and struggle to converge as effectively as those with conventional activations. This may indicate that their preserved plasticity comes from a reduced capacity to overfit, rather than from an inherently superior mechanism (like periodicity) for learning across tasks. In a sense, poorly fitting activation functions may act as a form of implicit regularization. While they currently do not offer a clear performance advantage, their ability to keep the network flexible remains interesting for potential future work.

## 7 Plasticity Preserving Training

Finally, we investigate whether lightweight training modifications can restore or preserve plasticity for activation-normalization pairs that otherwise degrade under class-incremental training. Concretely, we study (i) learning-rate warmup (Goyal et al., 2017) (+warmup) at the start of each task, (ii) label smoothing (+soft), and (iii) targeted L2 regularization (+tl2) that excludes affine parameters of normalization layers. Tables 5 (BatchNorm) and 6 (LayerNorm) summarize the final results (after task 20) for all activations that experienced measurable plasticity loss in prior sections. We argue that this three interventions target the same root cause that is over-commitment to the current task. Warmup reduces initial effective step sizes, soft labels bound the training confidence and delay loss minimization and excluding L2 from normalization affine parameters removes an unnecessary constraint on the model's capacity to re-center and re-scale features as the data distribution shifts.

**BatchNorm (Table 5)**. ReLU-family activations, which experience a significant plasticity loss in the vanilla No-Reset regime, recover strongly with warmup (and soft labels), reaching or surpassing their Reset baselines (e.g., ReLU: $72.06 \rightarrow 77.26$ with warmup+soft), while GELU benefits slightly from excluding L2 regularization from BatchNorm and SiLU remain essentially unchanged.

**LayerNorm (Table 6)**. Excluding L2 from LN affine parameters often improves plasticity and prevents collapse across activations (also demonstrated in Figure 6). In contrast to BatchNorm, we see a diverse improvement landscape w.r.t. training modifications. For ReLUs, adding warmup(+soft) recovers competitive accuracy (e.g., ReLU: $1.00 \rightarrow 74.07$), while SiLU, ELU, SELU etc. reach on-par or better when excluding L2 from LN is applied (and sometimes in combined with soft targets).

Table 5: **BatchNorm.** No-Reset runs that experienced plasticity loss are extended with small training modifications to evaluate their new performance after the final task.

| Activation | Reset Top-1 Acc. (%) | No-Reset | +warmup . . | +warmup +soft . | +warmup . +tl2 | +warmup +soft +tl2 |
|---|---|---|---|---|---|---|
| ReLU | 76.71 | 72.06 | 76.95 | **77.26** | 75.83 | 75.00 |
| Leaky-ReLU | 76.48 | 74.62 | 77.07 | **77.61** | 75.94 | 75.55 |
| GELU | 76.39 | 75.66 | 75.59 | 75.49 | **75.90** | 75.23 |
| ReLU-Decay | 76.26 | 70.46 | 76.76 | **77.20** | 74.00 | 74.90 |
| ReLU-Cut | 76.16 | 75.69 | 75.84 | **76.64** | 01.00 | 01.00 |
| SiLU | 75.95 | **75.55** | 75.49 | 75.53 | 75.24 | 75.46 |
| ReLU-Tanh | 75.85 | 66.13 | 75.78 | **76.36** | 71.05 | 71.67 |
| ReLU-Fade | 75.64 | 70.47 | 76.43 | **76.97** | 73.60 | 74.33 |

Table 6: **LayerNorm.** No-Reset runs that experienced plasticity loss are extended with small training modifications to evaluate their new performance after the final task.

| Activation | Reset Top-1 Acc. (%) | No-Reset | +warmup . . | +warmup +soft . | +warmup . +tl2 | +warmup +soft +tl2 |
|---|---|---|---|---|---|---|
| Leaky ReLU | 73.70 | 10.45 | 73.56 | **73.90** | 71.72 | 72.29 |
| ReLU | 73.48 | 01.00 | 73.68 | **74.07** | 71.38 | 72.37 |
| SiLU | 73.13 | 71.86 | 71.95 | 72.05 | 72.90 | **73.42** |
| ReLU-Tanh | 72.46 | **73.47** | 72.69 | 73.04 | 68.38 | 68.91 |
| ELU | 72.36 | 70.34 | 70.36 | 70.30 | **72.31** | 71.94 |
| ReLU-XY | 71.59 | 68.92 | 68.65 | 68.45 | **71.21** | 70.63 |
| SELU | 70.37 | 67.40 | 67.55 | 68.34 | 68.81 | **69.11** |
| GELU-Diff0 | 69.08 | 64.56 | 64.77 | 64.78 | 67.16 | **67.51** |
| ReLU-Y | 68.86 | 67.36 | 66.86 | 66.61 | **70.86** | 69.71 |
| GELU-Shift | 68.58 | 68.16 | 66.28 | 67.92 | 66.28 | **68.31** |
| Sine-Drift | 64.81 | **58.35** | 58.07 | 58.33 | 56.99 | 56.92 |
| HardTanh | 64.33 | 63.24 | 63.67 | **65.66** | 59.82 | 60.90 |

## 8  Recommendations for Deep Continual Learning

After conducting extensive comparisons and training across a wide range of activation-normalization pairs, we conclude by offering practical guidance for approaching continual learning problems. The following recommendations summarize our experience on how to design and train models that remain robust and adaptable for as long as possible.

**Key Insight.**  One of the central findings of this study is the critical role of regularization in any possible form to reduce overfitting during *non-stationary training phases*. This can be done implicitly, e.g. by choosing a sub-optimal performing activation function, or explicitly by soft labels, L2 regularization, learning rate warm up, normalization layers or other mechanisms.

**Practical Scenario.**  In some practical scenarios, where retraining from scratch is infeasible due to time or resource constraints, we recommend to intentionally over-regularize the model while the data distribution is not stable. Once stability is observed, either through statistical detection or expert judgment, a *model checkpoint* should be created. This checkpoint provides a safe fallback if future training is conducted. After checkpointing, the regularization strength can be reduced to allow a better adaptation to the (now stable)

data distribution. This strategy takes advantage of plasticity-oriented training when needed and shifts toward specialized, high-performance configuration when the environment is more stable.

**Practical Recommendations.**    Based on our empirical findings, we recommend the following architectural and training considerations for deep continual learning.

1. **Affine Parameters in Normalization Layers**

   Normalization layers (BatchNorm and LayerNorm) use learnable parameters.

   - Do not apply L2 regularization to the affine parameters.
   - This exclusion improves plasticity sometimes significantly. It allows stronger L2 regularization to be applied elsewhere in the network.

2. **Activation Functions**

   Activation functions impact both performance and adaptability.

   - Avoid *ReLU* and *LeakyReLU* in continual learning *unless* other forms of plasticity preservation are in place.
   - *GELU* and *SiLU* are good choices.
   - Try *ELU*, *ReLU-Cut*, *GELU-Core* or *ReLU-Y* with BatchNorm.
   - Try *ReLU-Tanh* or *GELU-Core* with LayerNorm.
   - Consider "weak" activation functions (e.g. *Tanh, SymSqrt* etc.) to implicitly regularize capacity. These may help prevent overfitting and preserve latent plasticity.

3. **Learning Rate Warmup**

   Warm-up phases can significantly improve adaptability for some activation functions.

   - In some real-world scenarios, these distribution shifts might not be easy to identify but considering the potential performance gains it is worth to try.

4. **Normalization Layers**

   We strongly recommend to use normalization layers.

   - Training runs with SELU activation show that its self-normalizing property is not sufficient to handle distribution shifts.

5. **Soft Labels**

   Apply soft labels as output regularization for long-term adaptability.

   - When training targets are continuous (e.g., in regression), consider reformulating them as classification with softened targets (2-hot encoding or HL-Gauss (Farebrother et al., 2024)).

6. **Overfitting**

   In general, do not let your model overfit to the current data distribution.

   - Experiment with other model regularization techniques when possible.

## 9    Limitations

Our results are reported on class-incremental CIFAR-100, as in Dohare et al. (2024), and track top-1 accuracy per task under Reset and No-Reset conditions. Results average over five seeds, and train 200 epochs per task. This work analyses supervised vision with simultaneous input and output shift and does not cover setups like RL, or large-scale regimes. Furthermore, all experiments use the backbone family ResNet-18, paired with BatchNorm, LayerNorm, or no normalization. Interaction effects between activations and normalization observed here may be specific to this residual CNN and need not transfer to deeper or wider ResNets, ConvNeXt/ViT-style models. Also, the study focuses on cross-entropy classification. Whether conclusions extend to contrastive or self-supervised objectives is unknown.

## 10 Conclusion

Plasticity loss presents a distinct and underexplored problem in continual learning, separate from catastrophic forgetting. In this work, we demonstrate that this phenomenon is strongly influenced by the interaction between architectural choices and the presence or absence of regularization and training routines. Through a large-scale empirical study spanning 26 activation functions and three normalization strategies, we demonstrate that the long-term adaptability of a network is heavily influenced by core architectural components. Rather than being easily interchangeable, activation functions and normalization layers interact in complex ways that significantly impact a model's ability to remain plastic. In addition, our findings challenge the assumption that performance and plasticity must inherently face a large trade off. In fact, we surprisingly showed that a ResNet-18 with ReLU + BatchNorm can stay plastic and maintain very strong performance which was previously (Dohare et al., 2024) unclear. Conversely, we found that overfitting correlates with decline in adaptability. Beyond architecture, we show that subtle training choices do matter very much. Excluding affine normalization parameters from L2 regularization, using soft labels, and introducing warmup schedules all improve the model's plasticity when distribution shift occur. Overall, our study shows there is no universal architecture solution for addressing plasticity in continual learning. The components interact in nuanced and often unpredictable ways which are also dependent on additional training schemes. Finally, we like to emphasize that the strategies we propose are not meant to replace existing active methods that attempt to fix plasticity loss. Rather, they are complementary. By proactively building models to remain plastic for longer, our recommendations extend the effectiveness of existing approaches and form a better foundation. However, continued investigation into this proactive approach is needed.

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

# A  Properties of Activation Functions

As part of our effort to identify plasticity patterns, we characterize each function according to a set of formal properties.

- **Left Asymptotic (LA):** A function $f(x)$ is called left asymptotic if as $x \to -\infty$, the function converges to a constant value $c$.

$$\lim_{x \to -\infty} f(x) = c. \tag{19}$$

- **Right Asymptotic (RA):** Analogously, $f(x)$ is said to be right asymptotic if as $x \to +\infty$, the function approaches a constant value $c$.

$$\lim_{x \to +\infty} f(x) = c. \tag{20}$$

- **Origin-Passing (O):** The function $f(x)$ is said to pass through the origin if it satisfies:

$$f(0) = 0. \tag{21}$$

- **Continuously Differentiable (CD):** A function $f(x)$ is continuously differentiable if

$$\lim_{\Delta x \to 0} f'(x + \Delta x) = f'(x), \quad \forall x \in \mathbb{R}. \tag{22}$$

- **Piecewise Linear (PL):** The function $f(x)$ has the property piece-wise linear (PL) if it only consists of linear segments, meaning that for each function-specific interval, the function has the form:

$$f(x) = m_i x + b_i \quad \text{for} \quad x \in [x_i, x_{i+1}] \tag{23}$$

Note that piece-wise linearity and continuous differentiability are not mutually exclusive. For example, the SELU activation function is neither PL nor CD.

- **Non-Negative (NN):** A function $f(x)$ is non-negative (NN) if it satisfies

$$f(x) \geq 0, \quad \forall x \in \mathbb{R}. \tag{24}$$

Non-negative functions (like ReLU or Softplus) can introduce optimization challenges in neural networks, particularly in gradient-based methods, by promoting "zig-zagging" behavior (Goodfellow et al., 2016b; LeCun et al., 2002).

# B  L2 Regularization

In this section, we present the (optimal) L2 regularization values used for each activation function under different normalization conditions: Batch Normalization, Layer Normalization, and no normalization. These values were set based on validation performance for the complete CIFAR-100 dataset during hyperparameter tuning.

Table 7: L2 regularization coefficients ($\lambda$) for each activation under three normalization regimes on CIFAR-100. A dash (—) denotes a combination that is not evaluated.

| Activation | BatchNorm | LayerNorm | No Norm |
|---|---|---|---|
| GELU-Core | 0.00070 | 0.00125 | — |
| ELU | 0.00025 | 0.00045 | 0.0001 |
| GELU | 0.00050 | 0.00100 | 0.0001 |
| HardSigmoid | 0.00008 | — | — |
| HardTanh | 0.00035 | 0.00020 | 0.0016 |
| LeakyReLU | 0.00050 | 0.00225 | 0.0010 |
| Log-Sigmoid | 0.00010 | 0.00002 | — |
| ReLU | 0.00050 | 0.00225 | 0.0010 |
| GELU-Diff0 | 0.00035 | 0.00070 | — |
| GELU-Shift | 0.00065 | 0.00145 | — |
| SELU | 0.00035 | 0.00030 | 0.0004 |
| ReLU-XY | 0.00025 | 0.00040 | — |
| ReLU-X | 0.00010 | 0.00010 | — |
| ReLU-Y | 0.00025 | 0.00035 | — |
| Sigmoid | 0.00010 | — | — |
| SiLU | 0.00035 | 0.00125 | 0.0001 |
| Sine | 0.00025 | 0.00280 | — |
| Sine-Drift | 0.00025 | 0.00007 | — |
| Softplus | 0.00010 | 0.00001 | — |
| ReLU-Cut | 0.00030 | 0.00100 | — |
| ReLU-Fade | 0.00060 | 0.00100 | — |
| ReLU-Decay | 0.00060 | 0.00100 | — |
| SymmetricLog | 0.00015 | 0.00025 | — |
| SymmetricSqrt | 0.00025 | 0.00035 | — |
| Tanh | 0.00025 | 0.00058 | 0.00085 |
| ReLU-Tanh | 0.00040 | 0.00200 | — |

## C   Model Architecture

This section provides a detailed specification of the ResNet-18 architecture (see Table 8) used for all experiments. The model is adapted to process $32 \times 32$ images from the CIFAR-100 dataset and always outputs predictions over 100 classes. We preserve the residual block structure while adjusting convolutional and downsampling configurations to suit the smaller image resolution. Each residual block consists of two convolutional layers followed by a normalization and activation layer, with downsampling performed via stride and shortcut connections when the feature map size changes.

Table 8: ResNet-18 architecture for CIFAR-100.

| Stage | Output Size | Layer(s) | Channels | Stride | Padding |
|---|---|---|---|---|---|
| Input | $32 \times 32 \times 3$ | — | — | — | — |
| Conv1 | $32 \times 32 \times 64$ | 3×3 conv, Norm, Act | $3 \rightarrow 64$ | 1 | 1 |
| Stage 1 (×2 blocks) | $32 \times 32 \times 64$ | 3×3 conv, Norm, Act
3×3 conv, Norm | $64 \rightarrow 64$ | $1, 1$ | $1, 1$ |
| Stage 2 (×2 blocks) | $16 \times 16 \times 128$ | 3×3 conv, Norm, Act
3×3 conv, Norm | $64 \rightarrow 128$ | $2, 1$ | $1, 1$ |
| Stage 3 (×2 blocks) | $8 \times 8 \times 256$ | 3×3 conv, Norm, Act
3×3 conv, Norm | $128 \rightarrow 256$ | $2, 1$ | $1, 1$ |
| Stage 4 (×2 blocks) | $4 \times 4 \times 512$ | 3×3 conv, Norm, Act
3×3 conv, Norm | $256 \rightarrow 512$ | $2, 1$ | $1, 1$ |
| Shortcut | varies | 1 ×1 conv (+ Norm) | in $\rightarrow$ out | same as block | 0 |
| Pool | $1 \times 1 \times 512$ | AdaptiveAvgPool2d | — | — | — |
| FC | 10 | Linear | $512 \rightarrow 10$ | — | — |

**Block:** Two 3×3 conv layers (first may downsample with stride 2), each followed by normalization and activation; shortcut is a 1×1 conv (plus Norm when changing channels); final activation after the residual sum.

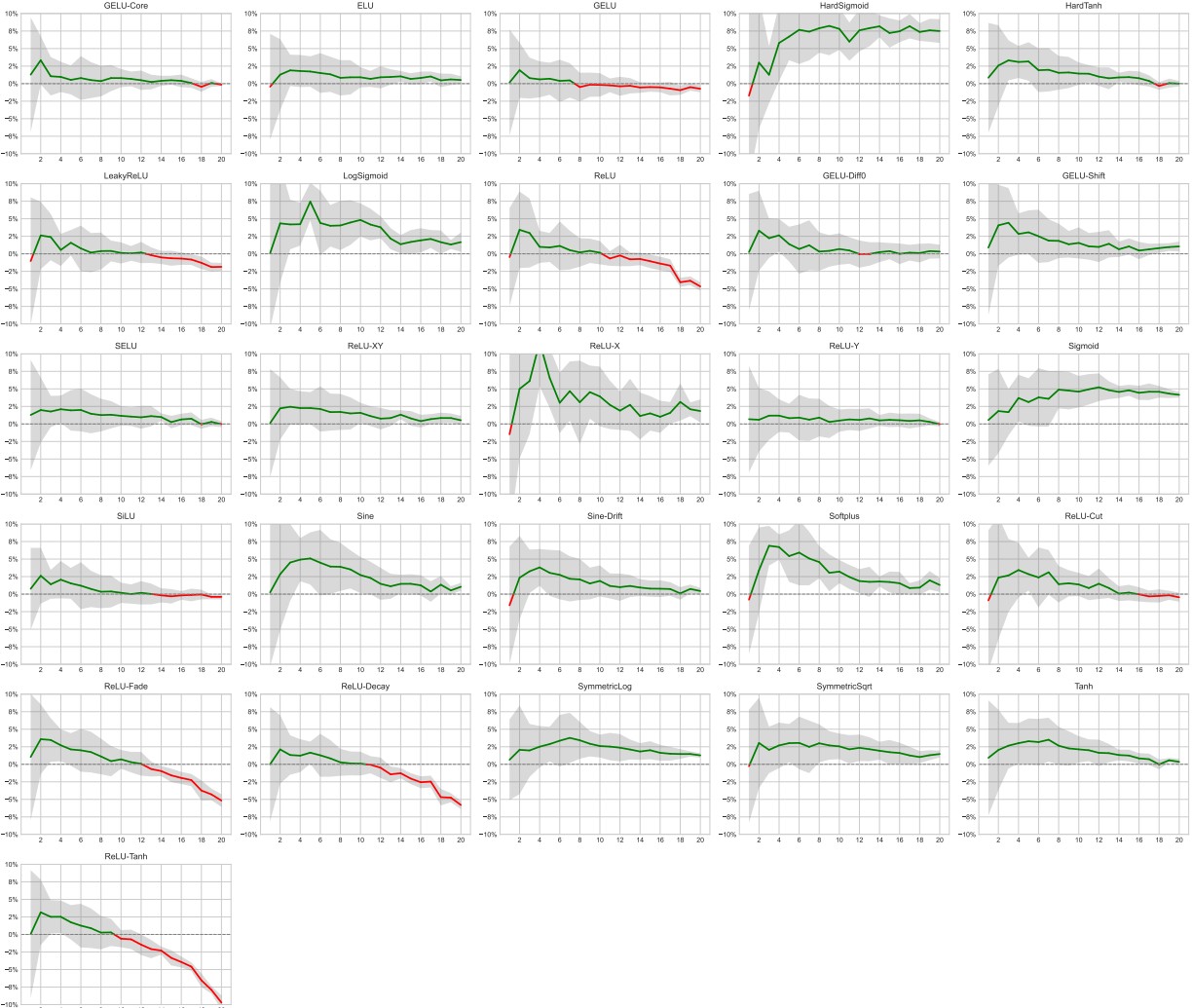

Figure 10: **BatchNorm: Comparison between *Reset* and *No-Reset*.** This figure compares test accuracies between *Reset* and *No-Reset* models using BatchNorm, based on the methodology shown in Figure 2. It consists of 26 subplots, each corresponding to a different activation function. The y-axis shows the absolute difference in test accuracy, with 0% representing the *Reset* model's test accuracy as the baseline. Values below 0% indicate a loss of plasticity (the *No-Reset* model underperforms the *Reset* model), while values above the baseline indicate improved performance. An improvement can happen because the *No-Reset* models are trained sequentially on every task.

## D  Additional Training Plots

This section presents additional information (Figure 10, 11, 12, 13, 14 & 15 ) that complements the main analysis. For each normalization strategy, we provide visualizations comparing the test accuracy difference between *Reset* and *No-Reset* training modes, as well as the relationship between final training loss and plasticity behavior. These figures extend the evaluation introduced in the main text and offer deeper insight into how optimization performance correlates with the preservation of plasticity.

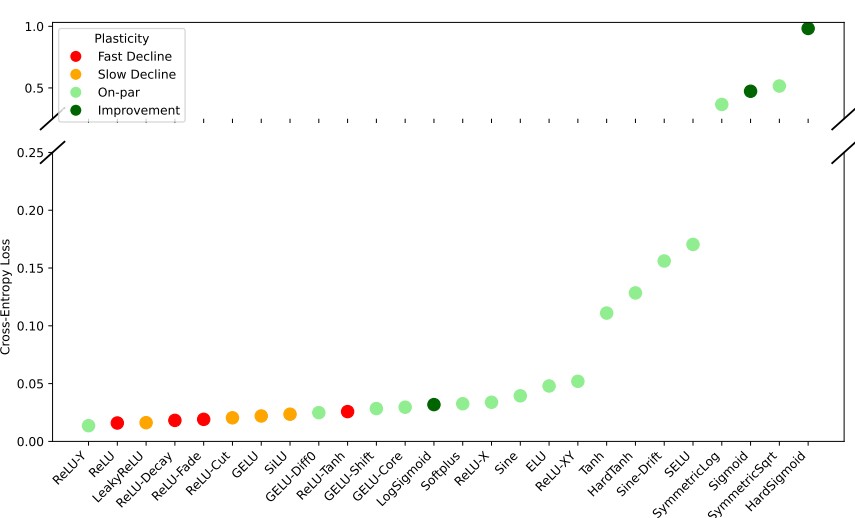

Figure 11: **BatchNorm: Relationship between Loss Minimization and Plasticity Behavior.**. This scatter plot shows how well different architectures minimize cross-entropy loss and how this relates to their plasticity under the *No-Reset* training. Each point represents a model trained from scratch on the full CIFAR-100, with the y-axis indicating the mean training loss over the final 200 steps (averaged across 5 seeds), and the x-axis denoting the architectural configuration. Point colors reflect plasticity behavior, as observed in Figure 10. The plot reveals: architectures that optimize training loss most aggressively often experience reduced plasticity.

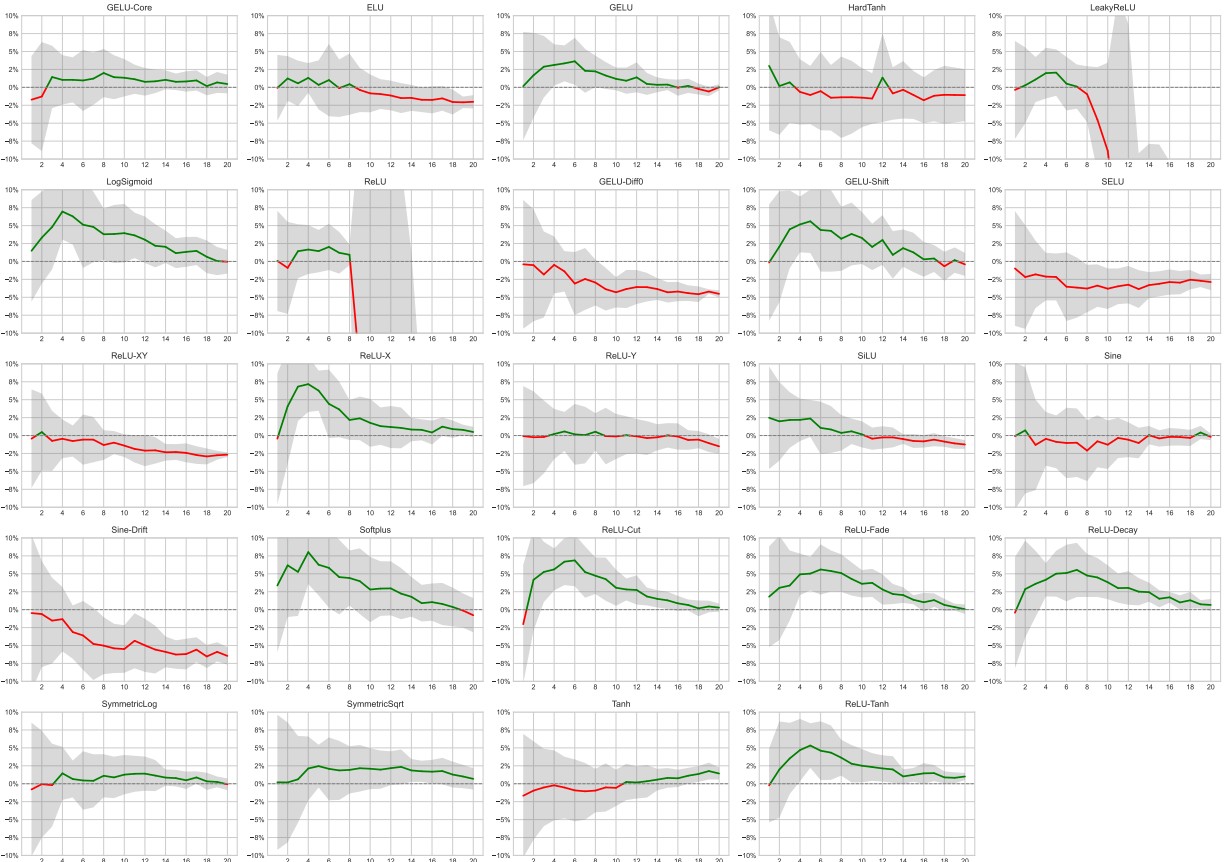

Figure 12: **LayerNorm: Comparison between *Reset* and *No-Reset* runs.** This figure compares test accuracies between *Reset* and *No-Reset* models using LayerNorm, based on the methodology shown in Figure 2. It consists of 24 subplots, each corresponding to a different activation function. The y-axis shows the absolute difference in test accuracy, with 0% representing the *Reset* model's test accuracy as the baseline. Values below 0% indicate a loss of plasticity (the *No-Reset* model underperforms the *Reset* model), while values above the baseline indicate improved performance. An improvement can happen because the *No-Reset* models are trained sequentially on each task.

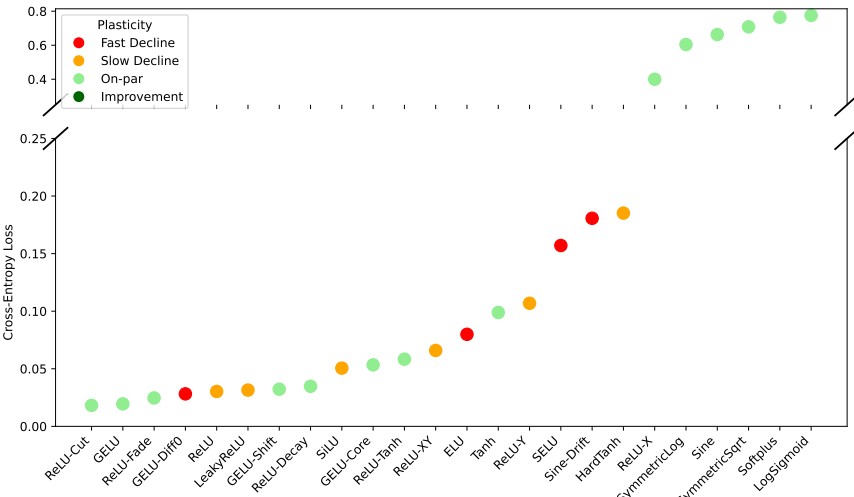

Figure 13: **LayerNorm: Relationship between Loss Minimization and Plasticity Behavior.** This scatter plot examines the relationship between training loss minimization and plasticity in models using Layer Normalization under the *No-Reset* training. Each point represents a model trained from scratch on the full CIFAR-100 dataset; the y-axis reports the mean cross-entropy loss over the final 200 training steps (averaged over 5 seeds), while the x-axis denotes the architectural variant. Point colors reflect plasticity behavior, as observed in Figure 12. For ReLU and LeakyReLU, we refer to Section 6 instead of the collapsed performance (see Table 1). The results reveal how architectural choices that favor loss minimization can sometimes impair long-term adaptability.

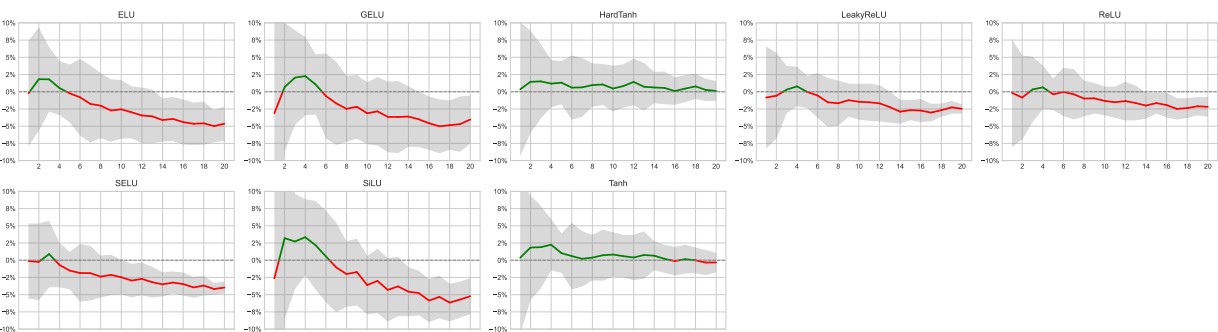

Figure 14: **No-Norm: Comparison between *Reset* and *No-Reset* runs.** This figure compares test accuracies between *Reset* and *No-Reset* models using No-Norm, based on the methodology shown in Figure 2. It consists of 8 subplots, each corresponding to a different activation function. The y-axis shows the absolute difference in test accuracy, with 0% representing the *Reset* model's test accuracy as the baseline. Values below 0% indicate a loss of plasticity (the *No-Reset* model underperforms the *Reset* model), while values above the baseline indicate improved performance. An improvement can happen because the *No-Reset* models are trained sequentially on each task.

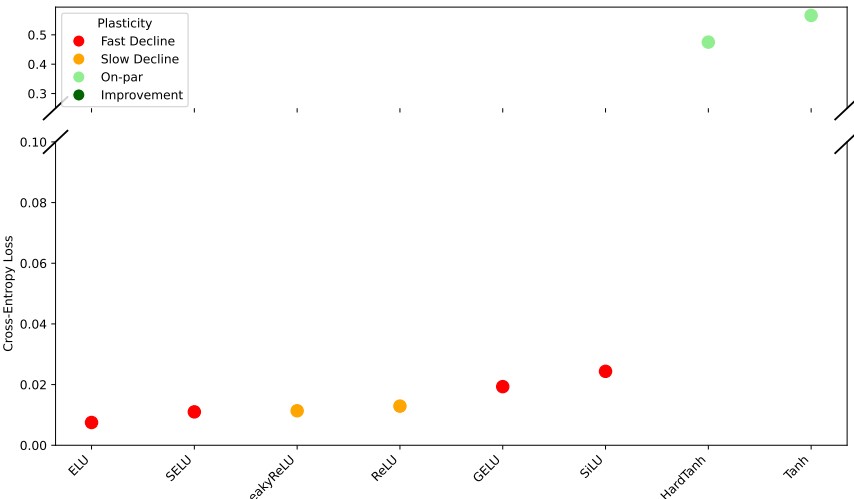

Figure 15: **No-Norm: Relationship between Loss Minimization and Plasticity Behavior**. This scatter plot analyzes the relationship between training loss minimization and plasticity for architectures without normalization, under the *No-Reset* protocol. Each point represents a model trained from scratch on the full CIFAR-100 dataset; the y-axis shows the mean cross-entropy loss over the final 200 training steps (averaged across 5 seeds), while the x-axis indicates the architectural configuration. Point colors denote plasticity behavior, as observed in Figure 14. The results emphasize that, in the absence of normalization, loss minimization alone does not guarantee adaptability.

## E   L2 Collapse with LayerNorm

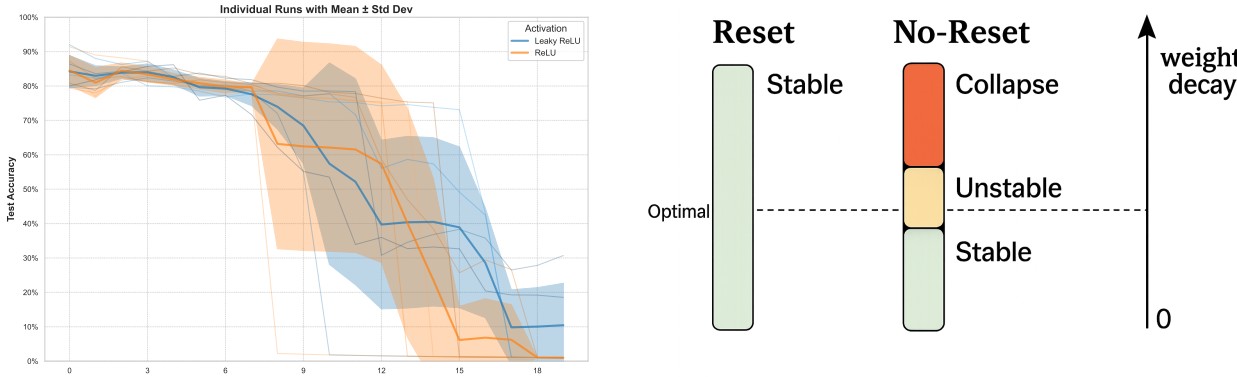

Figure 16: **Effect of L2 Regularization in Continual Learning with LayerNorm.** The left panel shows test accuracy performance across five runs each using ReLU (orange) and Leaky ReLU (blue), trained with optimized weight decay on CIFAR-100. After roughly nine tasks, all runs experience a significant performance drop. The right panel illustrates this *collapse*, defined as a consistent failure across all runs. We categorize outcomes as: **collapse** (all runs fail), **instability** (at least one fails), and **stability** (all remain stable). Reducing L2 regularization adequately prevents collapse and then also instability, but it sacrifices regularization strength for long-term adaptability.

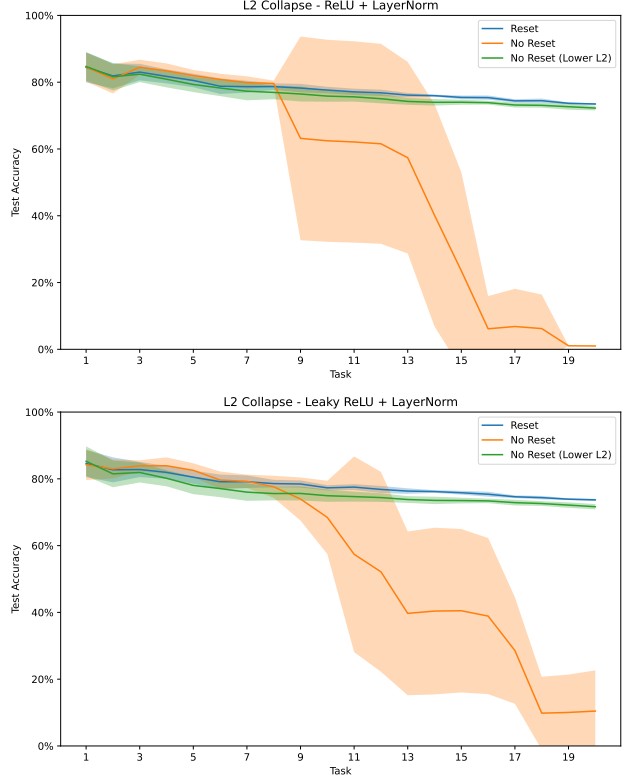

Figure 17: **Data on L2 Collapse supporting the Schematic in Figure 4.** The optimal L2 regularization (for the complete dataset) is not suited in continual learning. It needs to be reduced for stability.

## F  Learning Rate Schedules

To ensure good performance across diverse model configurations, we use step-wise learning rate schedules specific to each normalization strategy. Figure 18 illustrates the learning rate over 200 training epochs for models using BatchNorm, LayerNorm, and No-Norm. These schedules were selected based on empirical tuning and prior work Dohare et al. (2024), with early epochs using higher learning rates to accelerate convergence and later stages using gradual decay to get fine-tuning. The learning rate is updated at fixed epoch boundaries, and each schedule is designed to reflect the stability needs of the corresponding normalization strategy.

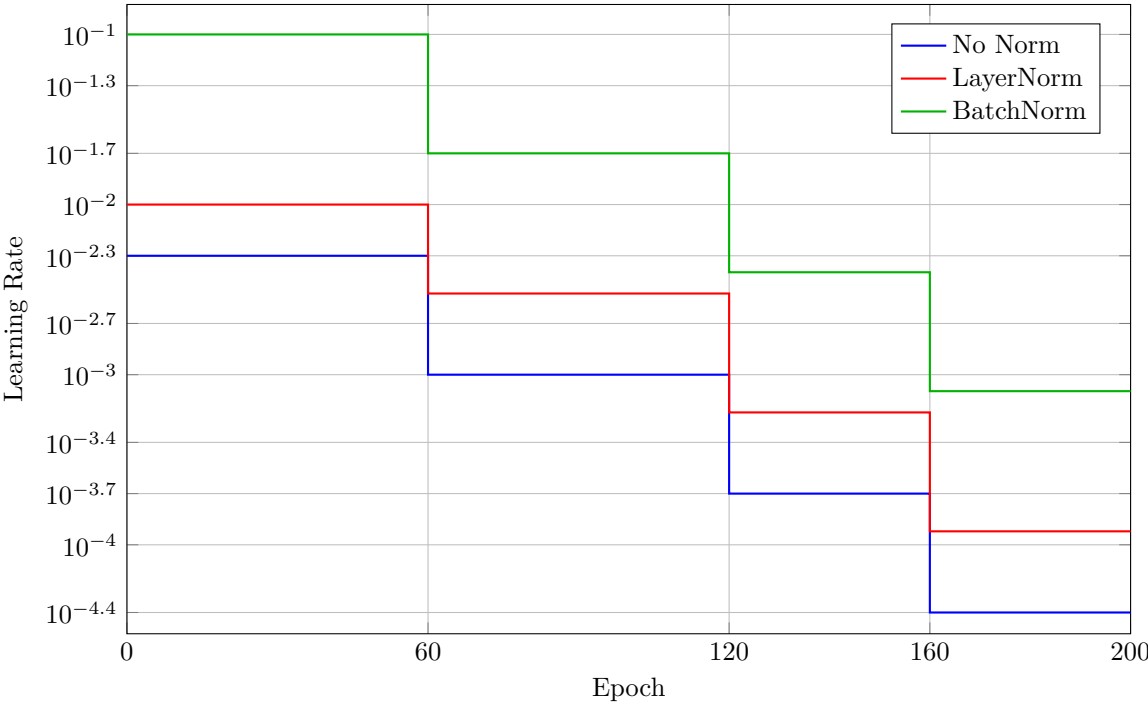

Figure 18: **SGD Learning Rate Schedules Across Normalization Strategies.** This figure shows the step-wise learning rate schedules used over 200 training epochs for models with BatchNorm, LayerNorm, and No-Norm. Each curve shows how the learning rate changes across epochs during the training on a single task.

# G   Code for Activation Functions

The last section provides the PyTorch implementation of all custom activation functions used in our experiments. These functions were designed to test a wide range of nonlinearities, including shifted, bounded, sinusoidal, and symmetric transformations. Each class follows a modular structure compatible with 'torch.nn.Module'. The code listings are grouped for clarity and include ReLU and GELU variants, symmetric functions, and activation functions with controlled saturation behavior.

```
1   import torch
2   import torch.nn as nn
3
4   class ReLU_X(nn.Module):
5     def forward(self, x):
6       return torch.max(torch.tensor(0.0), x + 1)
7
8   class ReLU_Y(nn.Module):
9     def forward(self, x):
10      return torch.max(torch.tensor(0.0), x) - 1
11
12  class ReLU_XY(nn.Module):
13    def forward(self, x):
14      return torch.max(torch.tensor(-1.0), x)
15
16  class GELU_Diff0(nn.Module):
17    def forward(self, x):
18      return torch.min(torch.tensor(0.0), torch.abs(x + 1.0) - 1.0) +
      torch.max(torch.tensor(0.0), x)
19
20  class GELU_Shift(nn.Module):
21    def forward(self, x):
22      return torch.min(torch.tensor(0.0), torch.abs(x + 0.5) - 0.75) +
      torch.max(torch.tensor(0.0), x)
23
24  class SineDrift(nn.Module):
25    def forward(self, x):
26      return x + torch.sin(x)
27
28  class SymSqrt(nn.Module):
29    def __init__(self, a=0.25):
30      super().__init__()
31      self.a = torch.tensor(a)
32
33    def forward(self, x):
34      return torch.sign(x) * (torch.sqrt(torch.abs(x) + self.a) - torch.sqrt(self.a))
35
36  class SymLog(nn.Module):
37    def forward(self, x):
38      return torch.sign(x) * torch.log(torch.abs(x) + 1)
```

Listing 1: Diverse custom activation functions

```
1
2   class ReLU_Tanh(nn.Module):
3     def forward(self, x):
4       return torch.max(torch.tensor(0.0), torch.tanh(x))
5
6   class ReLU_Cut(nn.Module):
7     def __init__(self, stop_value=1.0):
8       super().__init__()
9       self.stop_value = stop_value
10
11    def forward(self, x):
12      return torch.where(
13          x <= 0, torch.zeros_like(x),
14          torch.where(x <= self.stop_value, x, torch.zeros_like(x))
```

```
15           )
16
17    class ReLU_Fade(nn.Module):
18      def __init__(self, stop_value=1.0):
19        super().__init__()
20        self.stop_value = stop_value
21
22      def forward(self, x):
23        return torch.where(
24            x <= 0, torch.zeros_like(x),
25            torch.where(
26              x <= self.stop_value,
27              x,
28              torch.where(
29              x <= self.stop_value + 1,
30              self.stop_value * (1 - (x - self.stop_value) ** 2),
31              torch.zeros_like(x)
32              )
33            )
34          )
35
36    class ReLU_Decay(nn.Module):
37      def __init__(self, stop_value=1.0):
38        super().__init__()
39        self.stop_value = stop_value
40
41      def forward(self, x):
42        return torch.where(
43            x <= 0, torch.zeros_like(x),
44            torch.where(
45              x <= self.stop_value,
46              x,
47              self.stop_value / (1 + (x - self.stop_value) ** 2)
48            )
49          )
```

Listing 2: ReLU asymptote variants

```
1
2    class GELU_Core(nn.Module):
3      def __init__(self):
4        super(GELU-Core, self).__init__()
5
6        # Define breakpoints for the piecewise function
7        self.breakpoint1 = -2.0
8        self.breakpoint2 = -0.75
9        self.breakpoint3 = 0.0
10
11        self.m1 = -0.136    # Slope for Segment 2 (-2.0 <= x < -0.75)
12        self.b1 = -0.272    # Intercept for Segment 2
13        self.m2 = 0.227     # Slope for Segment 3 (-0.75 <= x < 0.0)
14        self.b2 = 0.0       # Intercept for Segment 3
15
16      def forward(self, x):
17        # Segment 1: x < -2.0 -> output = 0
18        seg1 = torch.zeros_like(x)
19
20        # Segment 2: -2.0 <= x < -0.75 -> output = m1 * x + b1
21        seg2 = self.m1 * x + self.b1
22
23        # Segment 3: -0.75 <= x < 0.0 -> output = m2 * x + b2
24        seg3 = self.m2 * x + self.b2
25
26        # Segment 4: x >= 0.0 -> output = x (identity)
27        seg4 = x
28
29        # Combine segments using nested torch.where
30        return torch.where(x < self.breakpoint1, seg1,
```

```
31                    torch.where(x < self.breakpoint2, seg2,
32                      torch.where(x < self.breakpoint3, seg3, seg4)))
33
```

Listing 3: GELU-Core activation

