# OpenReview forum: "Activation Functions and Normalization in Deep Continual Learning"
_TMLR — Rejected by TMLR_

### Review · Reviewer_nyZc · 2026-02-17

**Summary Of Contributions:**

The paper presents a study on the plasticity loss in continual learning. The study shows that the choice of the activation function and the normalization layer are contributing factors to the plasticity loss. However, no clear and general relationship emerges. The authors also explore variations of existing activation functions to identify characteristic behaviors. Based on the results of the empirical evaluation, the authors propose training recipes to mitigate the plasticity loss. The study is conducted with a ResNet-18 on the CIFAR-100 dataset.

The experiments are carried out in a rigorous way, and the results are nicely presented (with a few minor notes described later).
My main concerns are:
* The scope of the empirical results is very limited, as they only explore the CIFAR-100 dataset and the ResNet architecture. Since the main objective of the paper is to provide training recipes for continual learning, such recipes should strive to be as general as possible and to cover several continual learning setups. This is not the case for this paper.
* The plasticity loss is clearly present in the experiments (this was well-known from the literature). In most cases, the impact of all the empirical interventions is very limited. Also, the improved performance is observed at the very end of training, on the last 2-3 tasks. As an example, the result of a smoothing intervention is an increase of 0.5% in the test accuracy. This improvement is rather minimal. The same applies to the other interventions.
* The detailed analysis of each dimension of interest (batch norm, L2, learning rate scheduler, etc.) does not highlight common trends. Some configurations have something in common with others when considering a given dimension, but not never when considering all of them.
* The variations of the activation functions studied by the authors (e.g., ReLU and GELU) do not show a consistent behavior, with different variations possibly resulting in different degrees of performance-plasticity trade-offs.
* The final recipes proposed in the paper look too specific, tied to a case-by-case basis. I find it difficult to imagine how such recipes can help design continual learning pipelines, if not on CIFAR-100 with ResNet-18. The minimal advantages provided by the adjustments are not a convincing factor, either.

Overall, I think the objective of the paper is interesting and sound. Unfortunately, I struggle to find insightful results from this empirical evaluation.

**Audience:**

Yes

**Audience Explanation:**

Several continual learning papers have been published by TMLR, and continual learning is an active research topic in the machine learning community. The paper does not require a deep background to be understood, thus increasing the scope of the potential readers.

**Broader Impact Concerns:**

No concerns.

**Claims And Evidence:**

Yes

**Claims Explanation:**

The claims are sound and empirically validated.
While they are verified on the CIFAR-100 dataset, it is not possible to draw any conclusion on how much they will still hold for other datasets. The same reasoning applies to the neural architectures, as only ResNet-18 has been evaluated.

**Requested Changes:**

I do not have any specific adjustments in mind for the paper.
Extending the empirical evaluation is certainly a good starting point to improve the scope and the robustness of the results.
Carrying out more experiments could discover consistent relationships between the empirical dimensions considered in the paper. However, it is not possible to know the result of such experiments beforehand. From the current version, it does not look like such trends exist.

There are no experiments about the effect of the proposed adjustments on forgetting. It would be nice to see an analysis of this trade-off.

In terms of writing style, the paper is well-written and easy to read. One suggestion is about figure positioning, which is suboptimal at the moment. Often, figures are positioned a few pages before or after where they are referenced. Just as an example, I would suggest moving around figures 1, 2 and 3, as they are currently referenced a few pages after they appear. Also, Figure 2 is referenced in the text before Figure 1. This also applies to most of the figures in the results sections.

The test accuracy (y-axis) in most of the plots is not defined. In CL, that axis often corresponds to the accuracy on the entire test set of CIFAR-100. However, in the case of this paper, the accuracy is likely computed on the test set of the current training task, only. It would be better to clearly specify this in the text.

---

> ### Author Response · Authors · 2026-03-20
>
> We thank the reviewer for the detailed and constructive feedback!
>
> > The scope of the empirical results is very limited, as they only explore the CIFAR-100 dataset and the ResNet architecture.
>
> Please see our general comment.
>
> > Since the main objective of the paper is to provide training recipes for continual learning, such recipes should strive to be as general as possible and to cover several continual learning setups.
>
> Our main goal was not to claim universally applicable training "recipes" but to provide a controlled and systematic study of activation-normalization interactions in deep continual learning. We later added additional investigations (Section 4.2-5) and training mechanisms (Section 7) to see if plasticity improves for all pairs that showed degradation. Therefore, Section 8 (the "recipes") summarizes the findings from the previous chapters.
>
> > The plasticity loss is clearly present in the experiments (this was well-known from the literature). In most cases, the impact of all the empirical interventions is very limited. Also, the improved performance is observed at the very end of training, on the last 2-3 tasks. As an example, the result of a smoothing intervention is an increase of 0.5% in the test accuracy. This improvement is rather minimal. The same applies to the other interventions.
>
> We agree that the paper does not present a silver bullet that removes plasticity loss entirely, and we do not want to overclaim that. Our point is modest and it does not include mechanisms that are specifically designed for addressing plasticity loss (as in [1,2,3,4,5,6,7,8], see general comment). Simple design and training choices can delay the emergence of plasticity loss, reduce its severity, and sometimes recover a meaningful part of the gap across activation and normalization pairs. You simply get an improvement in test accuracy (**0.5% and up to 5%+**, see Table 6/7, Figure 5/6/7) for free. Why is this not valuable? Why is recovering 0.5% (and more) test accuracy considered insignificant when there were times when entire papers improved less than 0.5% on ImageNet?
>
>  Also note, that two tasks still represent 400 training epochs.
>
>  > The detailed analysis of each dimension of interest (batch norm, L2, learning rate scheduler, etc.) does not highlight common trends. Some configurations have something in common with others when considering a given dimension, but not never when considering all of them.
>
>  We partly agree, but we see this less as a weakness of the study and more as one of its key findings. The paper does not conclude that a single factor explains plasticity loss in general. In fact, one of the central messages is precisely that plasticity is influenced by interactions, not by one isolated design choice.
>
>  > The variations of the activation functions studied by the authors (e.g., ReLU and GELU) do not show a consistent behavior, with different variations possibly resulting in different degrees of performance-plasticity trade-offs.
>
>  Yes, that is true, but this is also exactly why we find these experiments informative. E.g. the activation-modification section is not meant to claim that a single simple property such as smoothness, saturation, or shifting the hinge fully determines plasticity. Rather, it investigates these hypotheses and shows that the answer is more complex. We would therefore frame this as a useful negative result.
>
>  > The final recipes proposed in the paper look too specific, tied to a case-by-case basis. I find it difficult to imagine how such recipes can help design continual learning pipelines, if not on CIFAR-100 with ResNet-18.
>
>  We did not frame Section 8 as "recipes". We carefully proposed recommendations, guidance and things to try based on the results.
>
>  > The minimal advantages provided by the adjustments are not a convincing factor, either.
>
> We have difficulty following this argument since a substantial number of models did improve their test set accuracy by 2-5% or sometimes more (see Table 5/6).
>
> > Extending the empirical evaluation is certainly a good starting point to improve the scope and the robustness of the results. Carrying out more experiments could discover consistent relationships between the empirical dimensions considered in the paper.
>
> As stated in the general comment, we are happy to work on additional experiments with datasets and/or models, but we would like to get some feedback on that.
>
> > There are no experiments about the effect of the proposed adjustments on forgetting. It would be nice to see an analysis of this trade-off.
>
> Good suggestion, but our work is positioned within a line of related work. As far as we know, catastrophic forgetting and loss of plasticity can be investigated independently from each other.
>
> > One suggestion is about figure positioning [...] It would be better to clearly specify this in the text.
>
> Thanks for the suggestions. We will fix this.
>
> Again, we appreciate the feedback. Thank you very much.

---

> > ### Comment · Reviewer_nyZc · 2026-03-23
> >
> > I thank the authors for the point-by-point answer.
> >
> > While I understand the authors' reasoning for relying only on CIFAR-100 and ResNet, I find that this clearly reduces the impact of their conclusions.
> > To me, this is even more true when such conclusions are not meant to be "general" (Our main goal was not to claim universally applicable training "recipes").
> > In short, I would consider this paper interesting and useful in one of these 2 cases:
> > 1) the paper provides a broad empirical evaluation showing the effectiveness of specific design choices with respect to the plasticity loss. Such design choices would be identified as promising plug-ins to existing training protocols to mitigate plasticity loss. To my understanding, it is difficult to extract such results from the current version of this paper.
> > 2) the paper provides clear evidence of mechanisms mitigating plasticity loss on a single, real-world dataset. Such mechanisms could then be used by people working with that dataset. This paper does not fall into this category as CIFAR-100 is not used in a real-world context, but only as a benchmark.
> >
> > The claim of the authors is that an empirical evaluation on a single dataset (a benchmark one) can still provide useful insights. I do agree with this statement. However, I do not think that the resulting publication meets the standard of TMLR.
> >
> > The authors also asked for specific suggestions on which additional experiments to run (a reviewer did suggest some: ImageNet-subsets). It is very easy to find additional datasets from the continual learning literature, and I do not think that the other reviewers or I are looking for specific datasets. Simply, the claims in the paper should be validated more extensively.
> >
> > What I am looking for is some regularity or recurring patterns in the results of the experiments. It is not that relevant to know that on CIFAR-100 a specific activation function is better than another. It is relevant to know that i) the choice of the activation function is relevant for the plasticity loss (the current experiments partially show this, but only for CIFAR, which does not prove the statement), ii) specific families of activation functions are always more effective than others in a broad empirical evaluation. Those would be actionable insights that could be useful for the community.
> >
> > I hope this clarifies my view on the current contribution.

---

> > > ### Author Response · Authors · 2026-03-25
> > >
> > > Thanks again for your constructive clarification. As previously mentioned, we are happy to work on additional experiments to further strengthen the submission. We certainly plan to move towards case 1 and are starting to work on:
> > >
> > > - Backbones: ResNet-18 + ResNet-34 (bigger) + one smaller non-ResNet (PPO/DQN CNN encoder)
> > > - Activations: only from Table 1
> > > - Datasets: CIFAR-100 + Tiny-ImageNet (+ ImageNet Subset with large images)
> > > - MLP on Permuted/Split MNIST.
> > > - Ablate further the training mods on new models/datasets.
> > >
> > > Unfortunately, this plan will take a bit longer due to manuscript, setup and training time.

---

### Review · Reviewer_siUX · 2026-02-25

**Summary Of Contributions:**

This manuscript studies the interaction between specific regularization techniques with certain combinations and plasticity loss in continual learning. The authors find that simple tricks such as soft labels or warm-up works for reducing plasticity degradation. Extensive controlled experiments support the claim.

**Additional Comments:**

Overall, the direction itself is worthy of being investigated, as the choice in regularization techniques might affect significantly the performance. Nevertheless, I think the generalization of the findings should be further investigated.

**Audience:**

Yes

**Audience Explanation:**

The direction itself is worthy of being investigated, as the choice in regularization techniques might affect significantly the performance. In this regard, several researchers may find certain values in this study.

**Claims And Evidence:**

No

**Claims Explanation:**

I think the current version is not sufficient. Please see the Requested Changes below.

**Requested Changes:**

- The observation and conclusions are significantly focused on the ResNet-18 with CIFAR-100 experiments. These observations might be different for other backbones and other datasets. The authors should test their findings on others.
- Similarly, although extensive experiments were performed, the authors should provide a convincing theoretical explanation that supports the generalization to others.
- The connection between training loss and plasticity might be difficult to decouple. The improvement on training loss can be brought about by those regularization techniques with respect to improved optimization, rather than improvement on plasticity.
- Expression of g in Eq. 3 should be revised. If g could be addition or multiplication, g should be RxR→R and C=g(\phi_1(x), \phi_2(x)).
- Writing should be improved.
    - It would be better to explicitly set stop_value=3.0 as the authors used \tau=3 across all variants.
    - The choice of a=0.25 in the source code should be explained for the intention.
    - Are the GELU-Offset and GELU-Shift the same or different?
    - “but also seem to” → “but also seems to”
    - “this three interventions” → “these three interventions”
    - “the the other” → “the other”
    - “e.g. see” → “e.g., see”

---

> ### Author Response · Authors · 2026-03-20
>
> Thanks for the constructive feedback and for the positive comments on the relevance of the direction.
>
> > The observation and conclusions are significantly focused on the ResNet-18 with CIFAR-100 experiments. These observations might be different for other backbones and other datasets. The authors should test their findings on others.
>
> Please see our general comment above.
>
> > Similarly, although extensive experiments were performed, the authors should provide a convincing theoretical explanation that supports the generalization to others.
>
> Our paper is primarily an empirical study, and we do not intend to claim theorem-level generalization.
>
> > The connection between training loss and plasticity might be difficult to decouple. The improvement on training loss can be brought about by those regularization techniques with respect to improved optimization, rather than improvement on plasticity.
>
> Our intention is not to claim a clean causal relationship that lower training loss directly causes plasticity loss in every case. Rather, we observe an empirical correlation between training loss minimization and reduced adaptability in the No-Reset setting. At the same time, we believe the label-smoothing intervention gives supporting evidence that the story is not only about immediate optimization quality. In particular, when label smoothing is removed late in training, performance improves again (see Figure 7). Still, we agree that this point should be described more carefully.
>
> > Writing should be improved. [...]
>
> Thank you. We will correct the writing issues you noted.
>
> > Overall, the direction itself is worthy of being investigated, as the choice in regularization techniques might affect significantly the performance. Nevertheless, I think the generalization of the findings should be further investigated.
>
> Thanks. As stated, we use best practices from [1] and additionally search for the optimal L2 hyperparameter (see Appendix B) for every activation-normalization pair to get the best performance for each model. Please refer to our general comment above.
>
> ---
>
> ## References:
> [1] Dohare, Shibhansh, et al. "Loss of plasticity in deep continual learning." Nature 632.8026 (2024): 768-774.

---

### Review · Reviewer_E79k · 2026-03-13

**Summary Of Contributions:**

The paper provides a large-scale empirical analysis exploring how the interaction between activation functions and normalization strategies affects plasticity loss in deep continual learning. All evaluations are performed  using a ResNet-18 backbone on a class-incremental CIFAR-100 benchmark. The authors assert that a model's long-term adaptability is determined by complex interactions between architectural choices rather than isolated design decisions. The study additionally proposes new activation functions, including ReLU and GELU variants, and identifies activation-normalization pairs that preserve plasticity. An important finding reveals strong correlation between aggressive loss minimization and plasticity loss. To mitigate this effect, the authors propose training modifications like soft labels, learning rate warm-up, and excluding L2 regularization on affine normalization parameters.

**Audience:**

Yes

**Audience Explanation:**

I might have chosen "Maybe" if it were an option, but I lean towards believing that the main findings are of interest to at least some of the TMLR audience studying plasticity loss.

**Broader Impact Concerns:**

The paper does not include a Broader Impact Statement. Given the empirical nature of the study, I don't see any problems with a lack of discussion of broader impact.

**Claims And Evidence:**

Yes

**Claims Explanation:**

The empirical evaluation in this paper is **extensive**, covering 26 activation functions across three normalization strategies (BatchNorm, LayerNorm, and No-Norm). The study uses a standard 20-task class-incremental CIFAR-100 benchmark  and a **Reset vs. No-Reset framework** to isolate and measure plasticity loss.

While the scope of the study is broad, several findings are **consistent with existing literature** are thus unsurprising:

- **Baseline Performance:** That standard activations like **ReLU and GELU** remain the top performers in the "Reset" (standard supervised) configuration is expected, as these functions are well-optimized for static training.
- **Plasticity Trade-offs:** That saturating activations (such as Tanh or Sigmoid) preserve plasticity better is evident from their **reduced capacity to fit (and thus to overfit)**.
- **Normalization Benefits:** The conclusion that explicit normalization layers are necessary to maintain plasticity reinforces established deep learning practice.

The biggest shortcoming of the empirical evaluation, as also noted by the authors, is its limitation to only the ResNet-18 architecture and the CIFAR-100 CIL benchmark. ResNet-18 is a low-capacity backbone and CIFAR-100 is a tiny image benchmark (so, less to learn). It is unclear if the main conclusions the authors wish to draw and the practical recommendations remain valid on larger images or stronger backbones.

**Requested Changes:**

As mentioned above, I think confirmation of the main findings on larger image benchmarks (e.g. ImageNet-Subset at least) and using stronger backbones is absolutely required. I don't claim that **all** results should be reproduced, but at least some experiments confirming that the main conclusions and practical recommendations scale to more realistic scenarios are needed.

---

> ### Author Response · Authors · 2026-03-20
>
> Thank you for your assessment of the paper! Based on your review, the main concern seems to be scope and insights rather than soundness, so we would like to address these points more directly.
>
> > While the scope of the study is broad, several findings are consistent with existing literature and are thus unsurprising.
>
> We agree that some high-level findings are aligned with prior work.
> But is the paper therefore less valuable? Results that look unsurprising in hindsight often only become reliable after repeated validation. A good example is normalization. Its importance in deep learning is now standard, but that conclusion became evident precisely because it was established empirically and moreover **repeatedly**.
>
> > Baseline Performance: That standard activations like ReLU and GELU remain the top performers in the "Reset" configuration is expected, as these functions are well-optimized for static training.
>
> Yes, ReLU and GELU perform well in static settings as shown in previous works, but it is not our intention to highlight this fact. We need these as a baseline, because it anchors the comparison against the continual-learning setting. Without that reference, it would be unclear whether differences in No-Reset behavior reflect plasticity effects or simply poor supervised performance.
>
> > Plasticity Trade-offs: That saturating activations preserve plasticity better is evident from their reduced capacity to fit.
>
> We think this is exactly the kind of claim that benefits from careful empirical validation. In continual learning, there is often tension between short-term fit and long-term adaptability, but it is not obvious how strong that effect will be across many activations and normalization choices. E.g. given enough gradient steps, these "bad" or underfitting activations **sometimes** (not always) become better because the "good" ones lose plasticity (see Table 1). We believe this is an important detail.
>
> > Normalization Benefits: The conclusion that explicit normalization layers are necessary to maintain plasticity reinforces established deep learning practice.
>
> Again, the important point is not normalization in isolation, but its interaction with activation choice in the continual setting. Our results show that these components should not be evaluated independently when reasoning about plasticity loss (see Table 1 and Appendix D).
>
> > The biggest shortcoming of the empirical evaluation, as also noted by the authors, is its limitation to only the ResNet-18 architecture and the CIFAR-100 CIL benchmark. ResNet-18 is a low-capacity backbone and CIFAR-100 is a tiny image benchmark.
>
> Please see our general comment (Section 1).
>
> > I think confirmation of the main findings on larger image benchmarks (e.g. ImageNet-Subset at least) and using stronger backbones is absolutely required.
>
> Please see our general comment (Section 1 and 2).

---

### Author Response · Authors · 2026-03-20
**General Comment**

We thank all the reviewers for their constructive feedback!

In our general comment, we would like to address two shared concerns.

# 1. Limitation to only the ResNet-18 architecture and the class-incremental CIFAR-100 benchmark.

This is a fair concern, and in general we agree that larger-scale confirmation would strengthen the paper. However, we would like to better contextualize the current study to related literature.

In deep continual learning, class-incremental CIFAR-100 is not a toy benchmark. It remains one of the more established and relatively larger evaluation settings for studying plasticity loss [1,2,3,4,5,6,7,8]. Likewise, while ResNet-18 may appear modest relative to modern large-scale vision models, it is also a standard backbone in the literature, and in **many well-established continual-learning papers it is the largest/strongest architecture evaluated** [1,2,3,4,5,6,7,8]. We therefore believe the current setup is representative of common practice in the area, rather than being unusually small by field standards.

Moreover, although the backbone is fixed, the empirical grid is already large in the dimensions central to our question: 26 activations, 3 normalization strategies, and multiple training interventions.

# 2. (Larger) Image benchmarks and stronger/other backbones

We understand the motivation and are happy to work on additional experiments.
We currently have access to 4x A100 GPUs, but continual-learning runs are expensive and time-consuming (given our compute budget). ImageNet-scale experiments in particular require **long** training times, since "No-Reset" needs significantly more compute time than standard "Reset" training. So, we are willing to add experiments, but at the moment, the requested additions remain somewhat unclear to us. Since no specific datasets, backbones, or prior work were suggested, we would appreciate clearer guidance. What kind of additional model architectures should we consider? Which datasets are relevant? How to design the continual learning procedure with novel datasets? What kind of additional experiments are needed?

---

## References:
[1] Dohare, Shibhansh, et al. "Loss of plasticity in deep continual learning." Nature 632.8026 (2024): 768-774.

[2] Lyle, Clare, et al. "Disentangling the Causes of Plasticity Loss in Neural Networks." Conference on Lifelong Learning Agents. PMLR, 2025.

[3] Yan, Hongwei, et al. "Orchestrate latent expertise: Advancing online continual learning with multi-level supervision and reverse self-distillation." Proceedings of the IEEE/CVF Conference on Computer Vision and Pattern Recognition. 2024.

[4] Guo, Yiduo, Bing Liu, and Dongyan Zhao. "Dealing with cross-task class discrimination in online continual learning." Proceedings of the IEEE/CVF conference on computer vision and pattern recognition. 2023.

[5] Wei, Yujie, et al. "Online prototype learning for online continual learning." Proceedings of the IEEE/CVF international conference on computer vision. 2023.

[6] Arani, Elahe, Fahad Sarfraz, and Bahram Zonooz. "Learning fast, learning slow: A general continual learning method based on complementary learning system." arXiv preprint arXiv:2201.12604 (2022).

[7] Guo, Yiduo, Bing Liu, and Dongyan Zhao. "Online continual learning through mutual information maximization." International conference on machine learning. PMLR, 2022.

[8] Caccia, Lucas, et al. "New insights on reducing abrupt representation change in online continual learning." arXiv preprint arXiv:2104.05025 (2021).

---

### Decision · Action_Editor_rEPk · 2026-05-23

**Recommendation:** Reject

**Audience:**

Yes

**Audience Explanation:**

The topic is relevant to the TMLR audience, especially researchers working on continual learning, plasticity loss, architectural design, normalization, and regularization. The study includes a nontrivial empirical grid over activation functions, normalization strategies, and training interventions, and some observations, such as
- the interaction between normalization and activation choice,
- the effect of excluding affine normalization parameters from L2 regularization,

may be useful to readers interested in plasticity-preserving training. However, the current scope limits the broader impact and generality of the findings.

**Claims And Evidence:**

No

**Claims Explanation:**

The paper presents an interesting empirical study, and the reviewers generally agree that the reported experiments are sound within the tested setting. However, all reviewers independently identified the same unresolved limitation: the conclusions and recommendations are drawn from a single dataset, a single architecture family, a supervised vision setting, and one CL protocol. This concern remains unresolved despite the authors’ rebuttal explaining why CIFAR-100/ResNet-18 should not be considered a “toy” setup, and it limits confidence in the generalization of the results to other settings.

Since the paper’s main contribution is empirical and recommendation-oriented, this limitation significantly weakens the generality and archival value of the work. The rebuttal contextualizes the choice of setup but does not provide additional evidence that resolves this concern. Therefore, I recommend rejection in the current round, while encouraging the authors to resubmit with additional datasets/architecture/CL setup and a more calibrated framing of the claims.

**Resubmission Of Major Revision:**

The authors may consider submitting a major revision at a later time.